# LikePhys: Evaluating Intuitive Physics Understanding in Video Diffusion Models via Likelihood Preference

**Jianhao Yuan**[†]    **Fabio Pizzati**[‡]    **Francesco Pinto**[§]    **Lars Kunze**[†◇]    **Ivan Laptev**[‡]
**Paul Newman**[†]    **Philip Torr**[†]    **Daniele De Martini**[†]
[†]University of Oxford    [‡]MBZUAI    [§]University of Chicago    [◇]UWE Bristol

## Abstract

Intuitive physics understanding in video diffusion models plays an essential role in building general-purpose physically plausible world simulators, yet accurately evaluating such capacity remains a challenging task due to the difficulty in disentangling physics correctness from visual appearance in generation. To the end, we introduce *LikePhys*, a training-free method that evaluates intuitive physics in video diffusion models by distinguishing physically valid and impossible videos using the denoising objective as an ELBO-based likelihood surrogate on a curated dataset of valid-invalid pairs. By testing on our constructed benchmark of twelve scenarios spanning over four physics domains, we show that our evaluation metric, Plausibility Preference Error (PPE), demonstrates strong alignment with human preference, outperforming state-of-the-art evaluator baselines. We then systematically benchmark intuitive physics understanding in current video diffusion models. Our study further analyses how model design and inference settings affect intuitive physics understanding and highlights domain-specific capacity variations across physical laws. Empirical results show that, despite current models struggling with complex and chaotic dynamics, there is a clear trend of improvement in physics understanding as model capacity and inference settings scale[1].

## 1 Introduction

Video diffusion models (VDMs) (Brooks et al., 2024; Google DeepMind, 2025; Polyak et al., 2024) have achieved impressive results in producing visually compelling videos, but they still often generate physically implausible outputs (Bansal et al., 2024; Motamed et al., 2025; Yuan et al., 2025). Ensuring generative models learn the underlying physics that govern the dynamics of visual data is essential not only to improve the outputs' quality (Kang et al., 2024; Li et al., 2025), but also a essential pre-requisit for them to serve as reliable world models (LeCun, 2022; Ha & Schmidhuber, 2018) with applications in robotics (Yang et al., 2023) and autonomous driving (Hu et al., 2023; Zhao et al., 2025).

However, it is challenging to evaluate how visual generative models learn and internalise physics. A classic line of work on general vision models uses the violation-of-expectation paradigm (Spelke, 1985; Baillargeon et al., 1985), which frames intuitive physics understanding as the ability to judge the plausibility of observed events (Riochet et al., 2018; Weihs et al., 2022), when presented with unrealistic videos obtained in simulation. However, extending this paradigm to generative models remains challenging. More recent approaches rely on Vision Language Models (VLMs) for plausibility judgments (Bansal et al., 2024; Guo et al., 2025) and text-conditioned compliance measures (Meng et al., 2024a). Although these methods provide valuable insights, they usually fail to disentangle physics from visual appearance (Motamed et al., 2025) or introduce subjective biases (Wu & Aji, 2023) and interpretive variance (Gu et al., 2024) through VLM judgments, ultimately struggling to provide a grounded assessment of intuitive physics understanding.

In this paper we propose an alternative evaluation method named *LikePhys*, that leverages the VDMs' density estimation capacity (Li et al., 2023) rather than relying solely on generated outputs, as we show in Fig. 1. Inspired by the violation-of-expectation paradigm (Riochet et al., 2018), we start from the assumption that the capacity of a model to assign higher likelihood to physically-plausible visual sequences is correlated to its intuitive physics understanding. Then, we use a simulator to render paired

---

[1]Data and code available at project page: https://yuanjianhao508.github.io/LikePhys/

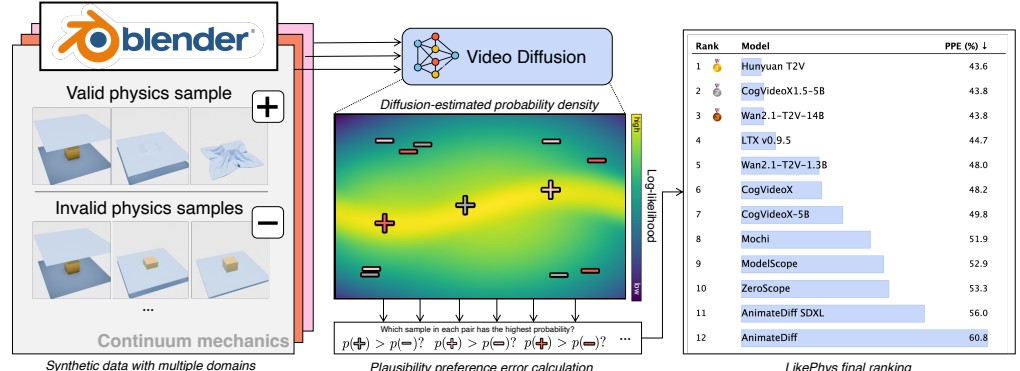

Figure 1: **Overview of *LikePhys*.** Our intuition is that a video diffusion model with a well learned underlying physics distribution should assign higher likelihoods to valid samples that obey physics laws and lower likelihoods to those invalid samples that violate them. We use Blender to create valid and invalid sample pairs through controlled physics parameters over multiple physics domains and scenarios (left). We then compare the diffusion model likelihood estimates over the constructed dataset to extract a quantitative intuitive physics understanding measure, the Plausibility Preference Error (PPE) (middle). Hence, we can compute a ranking of average PPE across pre-trained video diffusion models, that correlates with human preference. Lower values indicate stronger intuitive physics understanding (right).

videos. In one, we render physically-realistic phenomena, while in the other we introduce a controlled violation of physics. Importantly, we keep the visual appearance consistent in the pair, ensuring that any difference resulting from processing the videos can be attributed solely to the breach of physics principles. Both rendered videos are then corrupted with noise and processed by the diffusion denoising network, where noise prediction losses serve as a proxy for sample likelihood (Ho et al., 2020). We then calculate the Plausibility Preference Error (PPE), *i.e.* we assume positive scores if the physically plausible sample has higher likelihood, and negative otherwise. Aggregating PPE results across pairs yields a single score that quantifies the intuitive physics understanding of a single model.

To enable our evaluation, we construct a synthetic benchmark of twelve scenarios spanning Rigid Body Mechanics, Continuum Mechanics, Fluid Mechanics, and Optical Effects. Importantly, we design each physics scenario to be relatively simple, with clearly attributable governing physics dynamics, and consistent in appearance, so that any likelihood variation due to model preferences for visual quality cancels out during pairwise preference comparisons. Using this benchmark, we rank the performance of 12 state-of-the-art VDMs in terms of PPE, based on human preference-based verification. Furthermore, we analyse the key factors in model design and inference settings influencing their intuitive physics understanding, and examine domain-specific capacity variations across the physics domain and laws to highlight the limitations of current VDMs. Empirical results demonstrate that our proposed metric is a robust proxy for physics understanding in VDMs, offering actionable insights into both their current limitations and their potential for progress. Our key contributions are the following:

- We propose *LikePhys*, a training-free, likelihood-preference evaluation method for intuitive physics in VDMs with verified alignment to human preference.
- We benchmark pre-trained VDMs with a constructed dataset including twelve physics scenarios across rigid-body mechanics, continuum mechanics, fluid mechanics, and optical effects, each designed to isolate a specific physics violation under matched visual conditions.
- We conduct a comprehensive analysis on intuitive physics understanding of state-of-the-art VDMs, showing how architectural design and inference settings affect physics understanding, and highlighting variations in capacity across physics domains.

## 2 RELATED WORK

### 2.1 VIDEO DIFFUSION MODELS

VDMs have emerged as powerful generative frameworks for producing realistic video sequences (Esser et al., 2023; Brooks et al., 2024; Zhou et al., 2024; Girdhar et al., 2024). Early approaches

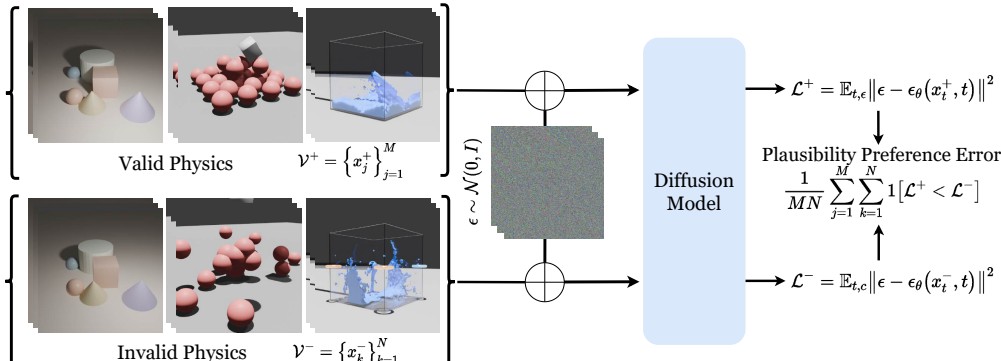

Figure 2: **Method Overview.** We prepare groups of videos via physics simulations with valid samples obeying physical laws and invalid samples containing deliberate violations. We then inject Gaussian noise into these videos and use a diffusion model to predict the noise and compute the denoising loss. For each valid–invalid pair, we compute a likelihood preference ratio that quantifies how the model favors physically plausible sequences, serving as a proxy for physics understanding.

such as VDM (Ho et al., 2022), AnimateDiff (Guo et al., 2024), and ModelScope (Wang et al., 2023) relied on 3D UNets (Ronneberger et al., 2015) to demonstrate the feasibility of spatio-temporal modeling. More recent systems, including CogVideoX (Yang et al., 2024), Hunyuan T2V (Kong et al., 2024), Wan (Wan et al., 2025), and LTX (HaCohen et al., 2024), adopt Transformer-based backbones in the Diffusion Transformer (DiT) style (Peebles & Xie, 2023; Vaswani et al., 2017), advancing long-sequence generation, visual quality, and inference efficiency. Despite these advances, it remains unclear whether current models capture underlying physical principles (Motamed et al., 2025). In this work, we propose a general evaluation protocol to assess the extent to which VDMs implicitly learn the laws of physics.

## 2.2 INTUITIVE PHYSICS UNDERSTANDING

Intuitive physics understanding is fundamental to a model's ability to reason about and predict scenes under physics laws (Ates et al., 2020; Bear et al., 2021; Zhan et al., 2024; Meng et al., 2024b). One important line of work builds on the violation-of-expectation paradigm (Spelke, 1985; Baillargeon et al., 1985; Margoni et al., 2024), framing physics understanding in general vision models as the ability to judge the plausibility of observed events (Smith et al., 2019; Riochet et al., 2020; Weihs et al., 2022; Garrido et al., 2025). For example, IntPhys1/2 (Riochet et al., 2018; Bordes et al., 2025) assess intuitive physics using tightly controlled synthetic video pairs that differ only by a single violation of permanence, immutability, spatio-temporal continuity, or solidity to study each physics principle in isolation. These works are mostly built on simulated data and have been successful in evaluating the physics understanding of various vision models (Garrido et al., 2025; Bordes et al., 2025). However, extending this protocol to visual generative models is non-trivial: it typically relies on context-conditioned generation and pixel reconstruction on target frames as a plausibility proxy, and it remains unclear how to adapt this setup to text-conditioned video diffusion models that lack image conditioning and are not naturally discriminative.

## 2.3 PHYSICS EVALUATION IN VIDEO GENERATIVE MODELS

Some recent works focus on evaluating video generative models. A popular branch of methods (Guo et al., 2025; Meng et al., 2024a) uses VLM with question–answering templates to assess adherence to physical laws in generated videos. For example, VideoPhy1/2 (Bansal et al., 2024; 2025) collect human-annotated synthetic videos and finetune VLM judges to align with human preferences on physical plausibility. However, these approaches are subject to visual appearance biases across different generative models (Wu & Aji, 2023) and interpretive variance across prompts and judging templates (Gu et al., 2024). Physics-IQ (Motamed et al., 2025) and Morpheus (Zhang et al., 2025) address these issues by comparing generated videos with paired real recordings via pixel/object-mask analyses and by fitting dynamical models to extract physical quantities, then scoring adherence to physical conservation laws, respectively. However, they rely on image-conditioned generation, and its extension to text-to-video generation also remains unclear.

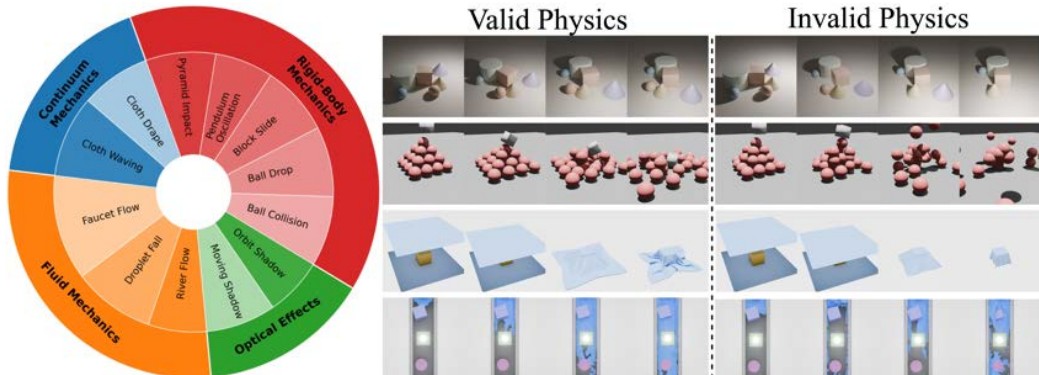

Figure 3: **Evaluation benchmark.** We organise 12 scenarios derived from four physical domains (Rigid Body Mechanics, Fluid Mechanics, Continuum Mechanics, Optical Effects) with their relative proportions (left). Rows from top to bottom (middle and right) show examples from four optical phenomena, rigid-body mechanics, continuum mechanics, and fluid mechanics. We display valid simulation (middle) and corresponding invalid variants (right) for physics violation in each domain.

To move beyond these limitations and provide an alternative evaluation from a likelihood preference perspective, we take inspiration from the violation-of-expectation paradigm while removing the need for conditional generation or pixel-level alignment. Our method adapts the density estimation capability (Li et al., 2023) of VDMs and performs pairwise comparisons by directly evaluating sample likelihoods through the denoising loss proxy. This design keeps the metric model-agnostic and appearance-agnostic, avoiding confounds from visual artifacts while focusing directly on physics understanding.

## 3 METHODOLOGY

We aim to evaluate the intuitive physics understanding of diffusion-based video generative models. We first define physics understanding from a distributional perspective, starting from preliminary assumptions (Section 3.1) and then deriving a likelihood preference score that quantifies the model's ability to assign a higher probability to physically valid samples than to invalid ones (Section 3.2). For the evaluation of *LikePhys*, we generate a comprehensive dataset of controlled video sequences using simulations (Section 3.3), which we use to test the sensitivity of the model to physical plausibility.

### 3.1 PRELIMINARIES ON VIDEO DIFFUSION MODELS

Diffusion Probabilistic Models (Sohl-Dickstein et al., 2015; Ho et al., 2020; Nichol & Dhariwal, 2021) learn the data distribution by inverting a forward Markov noising process: $q(x_t \mid x_{t-1}) = \mathcal{N}(x_t; \sqrt{1 - \beta_t}\, x_{t-1}, \beta_t I)$, $\beta_t \in (0, 1)$, A parameterised model then learns the reverse process $p_\theta(x_{t-1} \mid x_t) = \mathcal{N}(x_{t-1}; \mu_\theta(x_t, t), \sigma_t^2 I)$ by minimising a noise prediction loss which serves as an ELBO-based surrogate for the negative log-likelihood. The noise prediction loss is calculated by reconstructing the ground truth noise $\epsilon$ injected on $x_t$ with a denoising network $\epsilon_\theta$, following:

$$\mathcal{L}_{\text{denoise}}(\theta; x_t) = \mathbb{E}_{t, \epsilon} \left\| \epsilon - \epsilon_\theta(x_t, t) \right\|^2 \geq \mathbb{E}_{x_0} \left[ -\log p_\theta(x_0) \right] + \text{const.} \tag{1}$$

VDMs built on the framework that models dynamics in a sequence of frames $x_0 \in \mathbb{R}^{F \times C \times H \times W}$. The denoising network is therefore designed to capture both spatial structure within each frame and temporal relationship across frames, allowing the model to implicitly learn motion dynamics in addition to appearance.

### 3.2 PHYSICS UNDERSTANDING AS LIKELIHOOD PREFERENCE

Our intuition is to formalise intuitive physics understanding in a distributional perspective. Let $p_{\text{phys}}(x)$ denote the distribution over videos that strictly obey physical laws, where $x$ is a video sequence. We define its support as $\mathcal{M}_{\text{phys}} = \{ x \in \mathcal{X} : p_{\text{phys}}(x) > 0 \}$. By definition, any valid

sample $x^+ \in \mathcal{M}_{\text{phys}}$ satisfies real-world physical laws, whereas any invalid sample $x^- \notin \mathcal{M}_{\text{phys}}$ violates some of them. We consider a diffusion model $p_\theta$ that learn has a perfect intuitive physics understanding if, for every valid–invalid pair $(x^+, x^-)$,

$$p_\theta(x^+) \; > \; p_\theta(x^-). \tag{2}$$

Hence, if a diffusion model has learned the distribution correctly, it should assign a higher likelihood to valid samples when they are paired with visually matched invalid samples. As shown in Eq. (1), the denoising loss is an ELBO-based likelihood surrogate. Thus, a lower denoising loss corresponds to a higher likelihood under $p_\theta$. For any valid–invalid pair $(x^+, x^-)$, we therefore have

$$p_\theta(x^+) > p_\theta(x^-) \iff \mathcal{L}_{\text{denoise}}(\theta; x^+) < \mathcal{L}_{\text{denoise}}(\theta; x^-). \tag{3}$$

To evaluate this likelihood–preference protocol in practice, we first identify 12 physics scenarios. For each scenario, we create $R$ controlled variations to account for variability in both physics parameters and confounding visual factors (with $R = 10$ in our benchmark; see Sec. 3.3). For each variation $r \in \{1, \ldots, R\}$, we collect sets of controlled video pairs following the violation-of-expectation paradigm (Margoni et al., 2024; Bordes et al., 2025):

$$\mathcal{V}_r^+ = \{x_{r,j}^+\}_{j=1}^{M_r}, \qquad \mathcal{V}_r^- = \{x_{r,k}^-\}_{k=1}^{N_r}, \tag{4}$$

where $\mathcal{V}_r^+$ and $\mathcal{V}_r^-$ denote valid and invalid samples for variation $r$, respectively. For all video samples in $\mathcal{V}_r^+$ and $\mathcal{V}_r^-$, we ensure they are simple and consistent in visual appearance, with the only differences arising from violations of the governing physics dynamics. This ensures that any likelihood variation due to model preferences on visual style cancels out, enabling an unbiased comparison across different video diffusion models.

We then perturb both videos in each pair with the same Gaussian noise at sampled diffusion timesteps, pass them through the denoising network to predict the noise, and average the loss over timesteps as in Fig. 2, obtaining a denoising loss $\mathcal{L}_{\text{denoise}}(\theta; x)$ for each video. For a given variation $r$, we say the model has violated expectation on the pair $(x_{r,j}^+, x_{r,k}^-)$ if it assigns no greater likelihood (i.e., no lower denoising loss) to the valid sample than to the invalid one. Aggregating over all $N \times M$ pairs within each variation and then averaging across variations, we define the Plausibility Preference Error (PPE):

$$\text{PPE} = \frac{1}{R} \sum_{r=1}^{R} \frac{1}{M_r N_r} \sum_{j=1}^{M_r} \sum_{k=1}^{N_r} \mathbf{1}\Big[\mathcal{L}_{\text{denoise}}(\theta; x_{r,j}^+) \geq \mathcal{L}_{\text{denoise}}(\theta; x_{r,k}^-)\Big], \tag{5}$$

where $\mathbf{1}[\cdot]$ is the indicator function and the inner average corresponds to the rate that a model mis-assign higher likelihood to invalid video sample for each variation $r$. A lower PPE indicates that $p_\theta$ consistently prefers valid samples across variations, thus measuring the model's likelihood preference for physically plausible videos and serving as a proxy for its learned intuitive physics.

### 3.3 Evaluation Benchmark Construction

To rigorously evaluate the learned physics, we require controlled video pairs that differ only in physical validity. In practice, it is infeasible to obtain such matched pairs from real-world data, where physics laws cannot be systematically violated. We therefore construct a synthetic simulation benchmark spanning 12 physical scenarios across 4 domains, which allows precise control over physics parameters. All videos are rendered in Blender (Community, 2018) at $512 \times 512$ resolution with 60 frames. Each scenario is generated with $R = 10$ variations that vary both physics parameters and visual factors (e.g., object shape, texture, or environment). Within each variation, we generate $M$ valid videos that strictly obey the relevant law, and $N$ invalid variants that introduce a single controlled violation (e.g., superelastic bounces or ghosted shadows). By holding camera angle, lighting, textures, and object geometry constant within a single variation, the valid and invalid videos differ only in physical plausibility, ensuring that any measured likelihood gap can be attributed to physics violations. As shown in Fig. 3, the benchmark composition is as follows (see Apx. B for more details):

**Rigid Body Mechanics:** five cases including *Ball Collision*, *Ball Drop*, *Block Slide*, *Pendulum Oscillation*, and *Pyramid Impact*. They cover collision dynamics, periodic motion, gravity, and energy transfer. Valid examples conserve momentum and energy, follow free-fall under gravity, and

Table 1: **Model ranking.** Plausibility Preference Error (%) across twelve controlled physics scenarios for various video diffusion models. Lower values indicate stronger physics understanding. Models are ordered by *increasing* average performance (Avg) across all scenarios. Best and second-best per scenario are **bold** and underlined.

| | Rigid Body Mechanics | | | | | Cont. Mechanics | | Fluid Mechanics | | | Optical Effects | | |
|---|---|---|---|---|---|---|---|---|---|---|---|---|---|
| | Ball Collision | Ball Drop | Block Slide | Pendulum Oscillation | Pyramid Impact | Cloth Drape | Cloth Waving | Faucet Flow | Droplet Fall | River Flow | Orbit Shadow | Moving Shadow | Avg. |
| AnimateDiff | 63.3 | 60.0 | 65.0 | 51.7 | 61.1 | 52.9 | 78.6 | 62.0 | 63.3 | 44.0 | 71.7 | 56.0 | 60.8 |
| AnimateDiff SDXL | 63.3 | 66.7 | 38.3 | 70.0 | 68.9 | 47.1 | 82.9 | 54.0 | 60.0 | 46.0 | 43.3 | 32.0 | 56.0 |
| ZeroScope | 53.3 | 55.0 | 46.7 | 58.3 | 73.3 | 38.6 | 84.3 | 46.0 | 61.7 | **40.0** | 40.0 | 42.0 | 53.3 |
| ModelScope | 51.7 | 53.3 | 46.7 | 66.7 | 78.9 | 40.0 | 84.3 | 40.0 | 61.7 | **40.0** | 33.3 | 38.0 | 52.9 |
| Mochi | 40.0 | 50.0 | 31.7 | 65.0 | 84.4 | **28.6** | 82.9 | 38.0 | 60.0 | 44.0 | 58.3 | 40.0 | 51.9 |
| CogVideoX–5B | 43.3 | 41.7 | 61.7 | 65.0 | 83.3 | 38.6 | 81.4 | 36.0 | 53.3 | **40.0** | 23.3 | **30.0** | 49.8 |
| CogVideoX-2B | 40.0 | **38.3** | 60.0 | 63.3 | 83.3 | 35.7 | 81.4 | 36.0 | 50.0 | **40.0** | 18.3 | 32.0 | 48.2 |
| Wan2.1-T2V-1.3B | 43.3 | 53.3 | 66.7 | 20.0 | 40.0 | 65.7 | 32.9 | 60.0 | 33.3 | 78.0 | 23.3 | 60.0 | 48.0 |
| LTX v0.9.5 | 36.7 | 58.3 | 35.0 | 68.3 | **18.9** | 47.1 | 48.6 | **32.0** | 28.3 | **40.0** | 75.0 | 48.0 | 44.7 |
| CogVideoX1.5–5B | 61.7 | 50.0 | 51.7 | 31.7 | 25.6 | 52.9 | **22.9** | 54.0 | 21.7 | 68.0 | 23.3 | 62.0 | 43.8 |
| Wan2.1-T2V-14B | 41.7 | 56.7 | 61.7 | **13.3** | 43.3 | 62.9 | 34.3 | 56.0 | **15.0** | 64.0 | **16.7** | 60.0 | 43.8 |
| Hunyuan T2V | **33.3** | 51.7 | **25.0** | 21.7 | 50.0 | 35.7 | 77.1 | 38.0 | 40.0 | 68.0 | 38.3 | 44.0 | **43.6** |

maintain shape and continuity. Invalid variants break these rules through anomalies such as excessive restitution, teleportation, or implausible energy transfer.

**Continuum Mechanics:** two cases including *Cloth Drape* and *Cloth Waving*. They capture material deformation under gravity and aerodynamic forces. Valid examples show natural folds and consistent surface behavior. Invalid variants disrupt realism through penetrations, distortions, or scrambled temporal sequences.

**Fluid Mechanics:** three cases including *Droplet Fall*, *Faucet Flow*, and *River Flow*. They address conservation of mass, viscosity, and surface tension. Valid examples generate continuous and plausible fluid patterns. Invalid variants introduce reversed flows, discontinuities, spontaneous mass changes, or temporal glitches.

**Optical Effects:** two cases including *Moving Shadow* and *Orbit Shadow*. They capture light–object interactions. Valid examples show smooth and consistent shadow behaviour. Invalid variants break physical plausibility by inverting, erasing, or misaligning shadows or by adding temporal inconsistencies.

## 4 EXPERIMENTS

We evaluate the intuitive physics understanding of pre-trained VDMs using *LikePhys* and perform a systematic analysis to answer the following key questions: **(1)** Which models exhibit the strongest intuitive physics as measured by PPE? **(2)** How well does PPE align with human preference in physics correctness of generation? **(3)** To what extent does PPE disentangle visual appearance from intuitive physics? **(4)** What model design and inference factors influence models' intuitive physics? **(5)** Across which physics domains and laws do current models fall short? **Settings:** We benchmark pre-trained text-to-video diffusion models including AnimateDiff-SDv1.5/SDXL (Guo et al., 2024), ModelScope/ZeroScope (Wang et al., 2023), Mochi (Team, 2024), CogVideoX-2B/5B (Yang et al., 2024), Wan2.1-1.3B/14B (Wan et al., 2025), Hunyuan T2V (Kong et al., 2024) and LTX (HaCohen et al., 2024). To maximise performance, we adopt each model's recommended inference settings from its official repository and resample the frame rate and spatial resolution of generated video samples to match. We then perform zero-shot evaluation using the proposed *LikePhys* protocol, measuring performance via the PPE metric defined in Eq. (5) and ranking models. For each physics scenario, we report the average error over all valid–invalid pairs. For the likelihood estimation, we use the same DDIM (Song et al., 2020) scheduler with 10 timesteps uniformly sampled through the noise scale. We use a fixed prompt template for each physics scenario. More details of the evaluation implementation (Apx. A), evaluation dataset construction and specification (Apx. B), inference settings (Apx. C), experiment settings (Apx. D), further applicability to other intuitive physics benchmarks (Apx. E) and visual samples (Apx. F) are provided in the appendix.

Table 2: **Correlation to Human Preference.** Kendall's $\tau$ correlation between Plausibility Preference Error and state-of-the-art physics evaluators to human annotation across physics scenarios.

| | Ball Collision | Ball Drop | Block Slide | Pendulum Oscillation | Pyramid Impact | Cloth Drape | Cloth Waving | Faucet Flow | Droplet Fall | River Flow | Orbit Shadow | Moving Shadow | Overall |
|---|---|---|---|---|---|---|---|---|---|---|---|---|---|
| VideoPhy | 20.0 | -11.0 | 26.7 | -13.8 | -20.9 | 57.6 | 15.3 | 26.3 | -6.4 | 40.8 | 47.5 | -9.6 | 38.9 |
| VideoPhy2 | -19.4 | -32.2 | -6.8 | -15.9 | -3.1 | 51.0 | 10.5 | -20.0 | -41.0 | 6.8 | 70.4 | -33.9 | -8.5 |
| Qwen2.5 VL | 39.4 | 59.0 | -25.4 | -19.4 | 14.2 | 49.5 | -15.2 | 29.1 | -22.2 | 26.9 | 45.2 | -33.9 | 33.3 |
| *LikePhys* | 34.4 | 42.6 | 24.3 | 15.0 | 55.9 | 15.4 | 0.0 | 51.0 | -8.7 | 16.8 | 36.4 | 12.1 | 44.4 |

## 4.1 MAIN RESULTS

**Model ranking**   In Tab. 1, we benchmark twelve VDMs and order them by increasing PPE. The results reveal large performance gaps across architectures. Early UNet-based models (Ronneberger et al., 2015) such as AnimateDiff and ZeroScope often record error rates above 50%, reflecting limited ability to distinguish physically valid videos from invalid ones. In contrast, more recent DiT-based designs (Peebles & Xie, 2023), including the top three performers Hunyuan T2V (43.6%), Wan2.1–T2V–14B (43.8%), and CogVideoX1.5–5B (43.8%), achieve substantially lower errors across multiple scenarios (see Sec. 4.2 for further discussion).

Moreover, while there is a clear trend of improvement in recent architectures, only a handful of models significantly achieve PPE lower than the 50% random-guess threshold in more than a few scenarios, potentially due to simplicity biases, where physically implausible interactions appear visually easier to model than valid ones (Shah et al., 2020; Geirhos et al., 2020). These results highlight that while modern VDMs start to internalise physical principles, their performance varies substantially across different physics domains (see Sec. 4.3 for further analysis), indicating significant room for progress in physically-accurate training of video generators.

**Alignment to Human Preference**   Moreover, we conduct a human study to assess how well PPE serves as a proxy and aligns with human judgments of physics plausibility in downstream video generation. Specifically, we perform text-to-video generation with model tested in Sec. 4.1, generating 120 samples per model using the prompts that describe the designed physics scenarios in Sec. 3.3. Following the VideoPhy annotation protocol (Bansal et al., 2025), human annotators rate each video on a 1–5 scale, where 1 indicates a severe violation of physics and 5 indicates strong consistency (see Apx. D.2 for details). We then derive a ranking of models from the aggregated human scores and compare it against the PPE-based ranking of *LikePhys*, measuring their agreement with Kendall's $\tau$. We note that the VLM-based rankings leverage the dataset of videos generated by the candidate models, whereas the PPE-based ranking is computed solely on our synthetic benchmark and does not use any downstream model-generated videos.

We further compare PPE against state-of-the-art automatic evaluators, including human-aligned VLM-based metrics such as VideoPhy (Bansal et al., 2024) and VideoPhy2 (Bansal et al., 2025), as well as Qwen 2.5 VL 7B-Instruct (Bai et al., 2025; Jang et al., 2025) as a general VLM prompted to evaluate physical correctness. We find that PPE achieves a *stronger overall correlation with human annotations*, demonstrating the competitiveness of our zero-shot, training-free approach. In particular, the resulting correlation of $\tau = 0.44$ indicates that models with lower PPE also tend to be judged by humans as more physically consistent.

**Disentanglement of Visual Appearance** While we verify the correlation between PPE and physics consistency of generated videos, we examine whether it also correlates with established visual quality metrics from VBench (Huang et al., 2024) reported in Tab. 3. By doing so, we aim to determine whether perceptual metrics already capture physical plausibility or represent an independent dimension of evaluation. As shown in the table, aesthetic quality has no significant correlation ($r = -0.05$) to PPE, indicating our metric targets physical correctness rather than aesthetic appeal. Subject consistency

Table 3: Pearson correlation between Plausibility Preference Error (PPE) and visual quality metrics.

| | Corr. w. − PPE |
|---|---|
| Aesthetic Quality | -0.05 |
| Subject Consistency | -0.01 |
| Background Consistency | -0.01 |
| Motion Smoothness | 0.15 |
| Temporal Flickering | 0.12 |

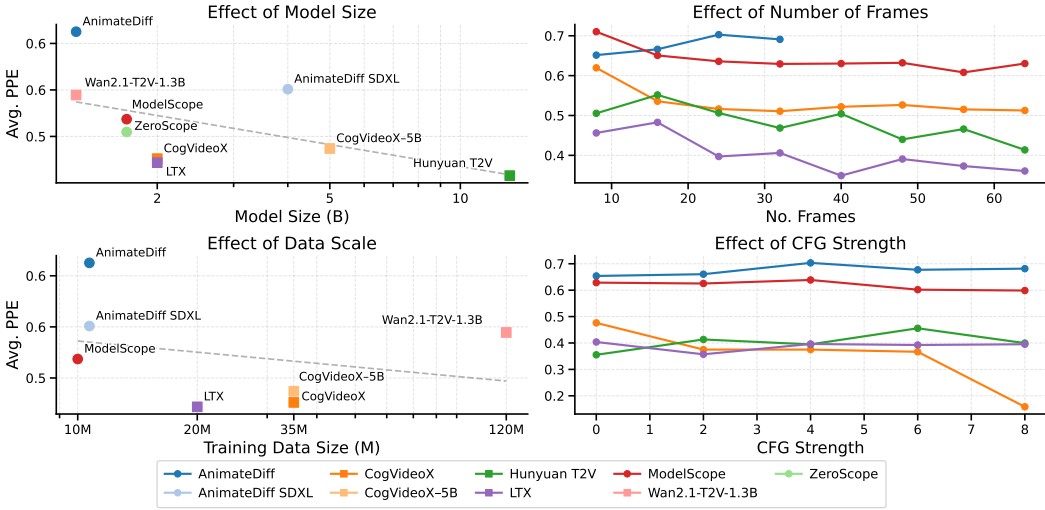

Figure 4: **Analysis on influencing factors.** Effect of model size, showing a steady overall decline in PPE *(top left)*. Effect of training data size, where larger corpora generally yield lower error, though the correlation is weaker than that of model size *(bottom left)*. Effect of context window size, showing consistent improvement in physics understanding as the window increases *(top right)*. Effect of classifier-free guidance (CFG) strength, indicating that physics understanding remains largely stable across different strengths *(bottom right)*.

($r = -0.01$) and background consistency ($r = -0.01$) show very weak correlations, consistent with the fact that the feature similarity-based metric does not fully capture physics correctness of dynamic motion. In contrast, motion smoothness ($r = 0.15$) and temporal flickering ($r = 0.12$) have moderate positive correlations with PPE. This is expected, as smooth, flicker-free motion can be a partial indicator of physical plausibility. Overall, these results confirm that our method evaluates aspects of physical reasoning that are largely orthogonal to visual quality measures, providing complementary insights into video generative models' capabilities and failure modes.

## 4.2 ANALYSIS ON INFLUENCING FACTORS

**Architecture and Data.** We analyse the physics understanding with respect to the scaling of model parameters and training data by plotting the average PPE in correlation to model parameters and training samples. As shown in Fig. 4, larger models consistently outperforms smaller variants. Largest models are almost exclusively DiT-based, showing the scalability of these architectures in capturing complex spatiotemporal dependencies. Overall, the average PPE also decreases with a larger training dataset size, indicating the effectiveness of architecture advancement and scaling to improve intuitive physics understanding. Please note that Hunyuan T2V does not disclose the training dataset cardinality.

**Role of Number of Frames.** Physical dynamics unfold over time, and the number of output frames is a critical design parameter for a video diffusion model's physics understanding. As shown in Fig. 4, most models exhibit a steady decrease in PPE as the number of frames increases. This trend suggests that providing longer temporal context allows the model to capture more complex interactions and improves its implicit physical reasoning.

**Role of Classifier-free Guidance.** Classifier-free guidance (CFG) (Ho & Salimans, 2022) is widely used to balance fidelity and diversity in diffusion sampling, but its effect on physics understanding remains unclear. We evaluate models across varying CFG strengths and find that guidance scale has only a marginal impact on physics understanding. This insensitivity suggests that, unlike visual quality metrics, physics plausibility is governed primarily by the learned model distribution, while inference-time calibration of noise prediction plays only a minor role. Consequently, CFG can be tuned for visual quality without compromising the model's physics understanding.

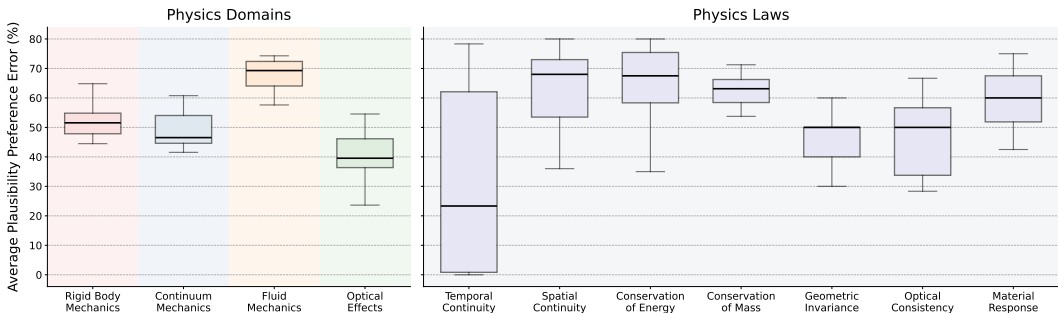

Figure 5: **Analysis across physics domains and laws.** On the left, we report detailed PPE across the four physics domains. Fluid Mechanics cases exhibit both the highest average error, while Rigid Body and Continuum Mechanics scenarios show moderate errors; Optical Effects cases lie in between. On the right, we map our domains to seven physics laws for a fine-grained analysis. Temporal continuity and conservation of energy show wide variation and higher median errors, whereas geometric invariance and optical consistency are handled more reliably.

## 4.3 Analysis Across Physics Domains and Laws

We investigate VDMs' physics understanding across different domains and laws presented in Sec. 3.3. As shown in Fig. 5, we report the average PPE across all models for each of the four proposed domains. Rigid Body and Continuum Mechanics show moderate errors with tight agreement across models, while Fluid Mechanics exhibits both higher median error and greater variability, where faucet flow or droplet fall scenes yield 30 to 40% PPE, but complex river flows often exceed 70%. Optical Effects incur the lowest errors, likely because large-scale image and video corpora strongly constrain photometric and geometric regularities. These findings indicate challenges in modelling nonlinear, multiscale dynamics.

We further analyse performance by physics law. Following the mapping in Apx. D.1, we group common invalid failure modes into seven laws and report the average PPE for each law. As shown in Fig. 5, temporal continuity shows the largest variance across models, indicating unstable long-range coherence when motion spans many frames or occlusions occur. Spatial continuity and conservation principles such as momentum and mass also yield high errors, which is consistent with the absence of global constraints in standard training objectives and samplers. In contrast, geometric invariance and optical consistency are better satisfied, likely because they align with priors learned from abundant static imagery and short clips. Material response remains moderately challenging, reflecting difficulty at contact events, frictional interactions, and surface compliance that require accurate high-frequency detail. Overall, models capture several structural and photometric regularities but still struggle with laws that require global coupling through time and space, suggesting future work on longer context training, multiscale memory, and physics-aware objectives that promote conservation and continuity.

## 4.4 Ablation on Evaluation Protocol Design

Beyond analysing model capacity, we also assess the robustness of our evaluation protocol by examining two key design factors that influence implicit physics understanding.

**Timestep Selection.** To accurately approximate the likelihood, we inject Gaussian noise at multiple timesteps. To adopt a proper sampling strategy, we investigate the denoising loss difference between valid and invalid sample pairs for four representative scenarios across the entire noise schedule. We found that (1) the overall preference trend is consistent across most timesteps for each model, and (2) the timesteps at which valid–invalid separation peaks vary by model and scenario. These observations imply that oversampling any single region could bias our estimate; therefore, we adopt uniform sampling of ten timesteps per paired comparison, striking a balance between computational efficiency and stable estimation.

**Prompt Robustness.** For text-to-video diffusion models, we also evaluate sensitivity to the choice of text prompt. For the same four scenarios, we create eight representative prompt variants. We then report the mean plausibility-preference error across prompts, with the standard deviation capturing

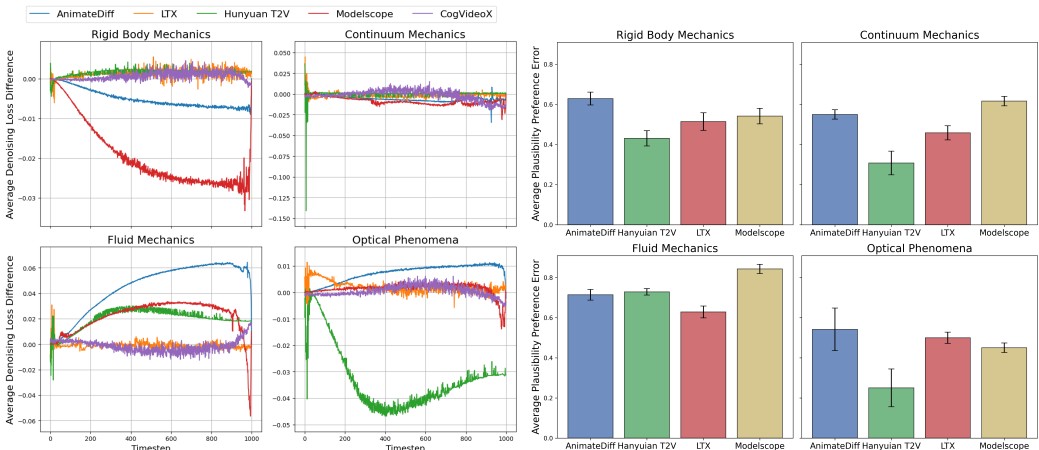

Figure 6: **Ablation on Evaluation Protocol Design Factors. Left**: Denoising loss differences between valid and invalid sample pairs across 1000 uniformly sampled timesteps for four representative physics scenarios. **Right**: PPE across multiple text-prompt variants, demonstrating robustness to prompt choice.

sensitivity to prompt choice. As shown in Fig. 6, we observe no significant change in discriminative performance when varying the prompt. This indicates that using the likelihood estimation from the text-conditioned distribution under different reasonable prompts would give similar performance. This robustness arises from our controlled data-generation process, which isolates only the physical-law compliance variable as the only error source.

## 5 DISCUSSION AND CONCLUSION

In this paper, we introduce *LikePhys*, a method for evaluating physical understanding in video diffusion models. By leveraging the violation-of-expectation paradigm together with the density estimation capability of video diffusion models, we ensure that any difference in denoising loss within a controlled video pair arises purely from physical implausibility. While our studies show the effectiveness of *LikePhys* in evaluating physics understanding, our approach also has limitations. We further discuss them and open new doors for research.

**Empirical Assessment of Physics Understanding.** Our method assesses whether the distribution learned by a video generative model is close to a physics-plausible distribution by comparing the likelihoods the model assigns to physically valid and invalid video pairs. Several factors can influence this physics understanding capacity, such as the training data distribution, suboptimal optimisation, and the training objective. However, we do not make any prior assumption about the cause of this capacity. Instead, *LikePhys* provides an empirical assessment of the resulting physics understanding.

**Cost of data curation.** One limitation of our method is that it relies on controlled physics violations, which is impossible to obtain in real-world. Extending to a broader range of scenarios would require additional data curation in simulator or video editing. While this is more costly compared to text-conditioned generation–based evaluation protocols, our approach offers the unique advantage of precisely controlling valid–invalid pairs to probe specific laws and failure patterns.

**Accessing Noise Prediction** Our method requires access to the noise prediction error of the model, which makes it difficult to evaluate closed-source models (Google DeepMind, 2025; Brooks et al., 2024). However, this requirement is less restrictive in the open source community. We believe that, during the development of closed-source models, our proposed method can be used as an indicator for *monitoring training progress* and *selecting checkpoints* during model release, which may require ranking hundreds of models.

ACKNOWLEDGMENTS

This work was supported by the EPSRC Programme Grant "From Sensing to Collaboration" (EP/V000748/1) and an RAi UK Enterprise Fellowship supported by EPSRC (EP/Y009800/1).

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

LLM USAGE

We use LLM for grammar polishing in paper writing. No text, code, or results were accepted without author verification. All technical content, claims, and conclusions are the authors' own.

## A DETAILS ON EVALUATION PROTOCOL

We provide more details on the implementation of our likelihood-preference evaluation protocol.

---

**Algorithm 1** Likelihood Preference Based Physics Understanding Evaluation

---

**Require:** Physics scenario subgroups, each with: $\mathcal{V}^+$ = set of $M$ physically valid videos $\mathcal{V}^-$ = set of $M$ physically invalid videos; Pretrained diffusion model $p_\theta$; Number of diffusion timesteps $T_{\max}$; Number of noise samples per timestep $N_\epsilon$;

**Ensure:** Plausibility Preference Error $PPE$

1: Initialize error count $E \leftarrow 0$
2: Initialize total comparisons $T \leftarrow 0$
3: **for** each physics subgroup **do**
4:     **Step 1:** Compute denoising losses for all videos
5:     **for** each video $x \in \mathcal{V}^+ \cup \mathcal{V}^-$ **do**
6:       $\mathcal{L}_{\text{denoise}}(x) \leftarrow 0$
7:       **for** $t = 1$ **to** $T_{\max}$ **do**
8:         **for** $n = 1$ **to** $N_\epsilon$ **do**
9:           Sample noise $\epsilon \sim \mathcal{N}(0, I)$
10:           Compute noisy frame

$$x_t \leftarrow \sqrt{\bar{\alpha}_t}\, x \,+\, \sqrt{1 - \bar{\alpha}_t}\, \epsilon$$

11:           Predict $\hat{\epsilon} \leftarrow \epsilon_\theta(x_t, t)$
12:           Accumulate

$$\mathcal{L}_{\text{denoise}}(x) \leftarrow \mathcal{L}_{\text{denoise}}(x) \,+\, \|\epsilon - \hat{\epsilon}\|_2^2$$

13:         **end for**
14:       **end for**
15:       Normalize:

$$\mathcal{L}_{\text{denoise}}(x) \leftarrow \frac{\mathcal{L}_{\text{denoise}}(x)}{T_{\max} \times N_\epsilon}$$

16:     **end for**
17:     **Step 2:** Pairwise comparisons between valid and invalid videos
18:     **for** each $x^+ \in \mathcal{V}^+$ **do**
19:       **for** each $x^- \in \mathcal{V}^-$ **do**
20:         $T \leftarrow T + 1$
21:         **if** $\mathcal{L}_{\text{denoise}}(x^+) > \mathcal{L}_{\text{denoise}}(x^-)$ **then**
22:           $E \leftarrow E + 1$ {model prefers invalid over valid}
23:         **end if**
24:       **end for**
25:     **end for**
26: **end for**
27: Compute error rate:

$$PPE \leftarrow \frac{E}{T}$$

28: **return** $PPE$

---

## B DETAILS ON BENCHMARK DATASET GENERATION

We provide details on each of the constructed physics scenarios, including the settings and the implementation of how the invalid cases are generated.

- **Ball Collision:** Two spheres roll toward each other and collide elastically on a flat surface, conserving momentum and energy. In invalid variants, restitution is altered (e.g. perfectly inelastic "sticking" or super-elastic energy amplification); one sphere penetrates the other at impact; phantom forces are applied during collision; sphere radii change mid-impact; one sphere teleports through the other; or collision timing is scrambled (temporal disorder).

- **Ball Drop:** A sphere drops from a fixed height onto a rigid floor and bounces back under gravitational energy conservation. In invalid variants, the sphere's color changes mid-flight; its radius is dynamically rescaled during free fall; it bounces higher than gravity permits ("over-bounce"); it penetrates the floor; it teleports to a different height; or its temporal sequence is disrupted (temporal disorder).

- **Block Slide:** A rectangular block slides down an inclined plane under friction and gravity, following Newton's laws. In invalid variants, the block hovers above the plane; it moves erratically, breaking Newtonian motion; spurious jitter is injected; its dimensions change as it slides; it teleports partway down the slope; or its time evolution is scrambled (temporal disorder).

- **Cloth Drape:** A rectangular cloth drapes over a cylindrical support, responding to gravity and bending forces while preserving surface continuity. In invalid variants, the cloth's color changes mid-simulation; it penetrates the cylinder or ground; it folds in impossible ways, violating material continuity; it behaves like a rigid sheet with no natural flutter; or its temporal evolution is scrambled (temporal disorder).

- **Cloth Waving:** A piece of cloth attached to a vertical pole flutters under a constant wind force, exhibiting realistic wave propagation and strain. In invalid variants, sections freeze instantly; fragments shatter mid-wave; parts teleport from one side to the other; an impossible 180° twist is imposed; it explodes outward as if under internal pressure; or its motion breaks into non-physical jump cuts (temporal disorder).

- **Pyramid Impact:** A cube drops onto a pyramid of stacked spheres, transferring energy and knocking spheres off under gravity. In invalid variants, collision energy is artificially amplified or damped (energy-conservation violation); holes appear in the pyramid (mass continuity violation); spheres or the cube teleport through each other (spatial discontinuity); or gravity is negated so objects float unpredictably.

- **Pendulum Oscillation:** A pendulum bob swings on a rigid rod under constant gravity, following a consistent periodic arc. In invalid variants, the rod breaks mid-swing (path discontinuity); the bob disappears for segments; its trajectory deviates from the circular path; time intermittently freezes (temporal disorder); or the length/frequency varies randomly (frequency variation).

- **Droplet Fall:** A single droplet falls under gravity, forming a coherent drop and splash while conserving mass and momentum. In invalid variants, antigravity forces make the droplet rise; the stream fragments into disconnected blobs; fluid mass is spontaneously created or removed (mass-conservation violation); viscosity becomes negative or oscillates; particles self-attract unnaturally; or its temporal sequence is scrambled (temporal disorder).

- **Faucet Flow:** A faucet pours water into a transparent tank, forming a continuous stream that collects and splashes, conserving mass and momentum. In invalid variants, fluid color shifts mid-flow; the stream fractures into non-coalescing droplets; viscosity becomes negative or oscillates; fluid mass is injected or removed arbitrarily (mass-conservation violation); phase shifts alternate liquid-gas instantly; particles self-attract unnaturally; droplets teleport across the tank; or temporal order is scrambled (temporal disorder).

- **River Flow:** Water flows steadily downstream along a riverbed, exhibiting realistic laminar or turbulent patterns under gravity. In invalid variants, the flow fractures into isolated droplets; an invisible barrier blocks water without visual cues; fluid mass vanishes mid-flow; phases shift liquid→solid→liquid; or timestamps jump (temporal disorder).

- **Moving Shadow:** A solid object moves across a ground plane under fixed illumination, producing a consistent, smoothly moving shadow. In invalid variants, the shadow inverts onto the ceiling; it vanishes entirely; it appears without its caster; its shape mismatches the object; it teleports away; or its temporal sequence is scrambled (temporal disorder).

- **Orbit Shadow:** A shadow orbits around an object under fixed lighting, tracing a smooth circular path. In invalid variants, the shadow inverts direction or plane; it vanishes mid-orbit; it detaches from its caster; its geometry distorts implausibly; it teleports along its path; or its temporal order is scrambled (temporal disorder).

## C  ADDITIONAL DETAILS OF INFERENCE SETTINGS

Our evaluation leverages off-the-shelf video diffusion pipelines in a zero-shot setting. For each model, we adopt the officially recommended spatial and temporal resolutions as shown in Tab. 4 and apply a uniform noise scheduler (Euler discrete) over a fixed number of timesteps ($T = 10$) as uniform sampling found optimal in Diffusion Classifier (Li et al., 2023).

Each input clip is first uniformly sampled to the prescribed frame count and resized to the model's native ($H \times W$) resolution. The resulting sequence is encoded into a latent representation by the model's VAE. To approximate the model's internal likelihood, we inject Gaussian noise at $T$ evenly spaced timesteps, run the noised latent through the diffusion backbone (with classifier-free guidance at the specified scale), and record the mean-squared error between the true noise and the network's prediction. Averaging these per-step errors yields a scalar loss that serves as a proxy for $-\log p_\theta(x)$.

For each physics scenario, valid and invalid variants are processed identically, and their losses are compared pairwise. The Plausibility Preference Error is computed as the fraction of valid–invalid pairs in which the invalid sample attains a lower denoising loss than the valid one. All sources of randomness (Python, NumPy, PyTorch, CUDA/cuDNN) are fixed for the video-pairs but different across subgroups via a global seed to ensure full reproducibility.

Table 4: Spatial and temporal settings for zero-shot inference.

| Model | Height | Width | FPS | #Frames |
|---|---|---|---|---|
| AnimateDiff | 512 | 512 | 16 | 16 |
| AnimateDiff SDXL | 1024 | 1024 | 16 | 16 |
| CogVideoX 2/5B | 480 | 720 | 16 | 49 |
| Zeroscope / ModelScope | 320 | 576 | 16 | 24 |
| Wan2.1-T2V (1.3B/14B) | 480 | 832 | 16 | 33 |
| Hunyuan (I2V / T2V) | 320 | 512 | 16 | 61 |
| LTX variants | 480 | 704 | 25 | 161 |

All the model evaluated are text-conditioned video diffusion models, we use a prompt template for each scenario that describes the physically valid event, and a shared negative prompt to discourage spurious artifacts: `"worst quality, inconsistent motion, blurry, jittery, distorted"`.

- **Ball Collision:** `"two balls colliding with each other"`
- **Ball Drop:** `"ball dropping and colliding with the ground, in empty background"`
- **Block Slide:** `"a block sliding on a slope"`
- **Pendulum Oscillation:** `"a pendulum swinging"`
- **Pyramid Impact:** `"a cube crash into a pile of spheres"`
- **Cloth Drape:** `"a piece of cloth dropping to the obstacle on the ground"`
- **Cloth Waving:** `"a piece of cloth waving in the wind"`
- **Droplet Fall:** `"a droplet falling"`
- **Faucet Flow:** `"fluid flowing from a faucet"`
- **River Flow:** `"fluid flowing in a tank with obstacles"`

- **Moving Shadow:** `"light source moving around an object showing its shadow"`

- **Orbit Shadow:** `"camera moving around an object"`

# D ADDITIONAL EXPERIMENTAL SETTING DETAILS

## D.1 CLASSIFICATION OF PHYSICS LAWS

We provide details on how we classify and summarise specific physics laws in Tab. 5.

## D.2 HUMAN STUDY EXPERIMENT DETAILS

For the human study experiment, we generate 10 samples per physics scenario per model using the same prompts as shown in Apx. C with prompt enhancement from GPT5. Human annotators are asked to rate each video on a scale of 1–5, following the annotation protocol of VideoPhy2 (Bansal et al., 2025), where higher scores indicate stronger physical consistency.

For baseline physics evaluator comparison, we leverage VideoPhy1 (Bansal et al., 2024), which rates videos on Physics Consistency (PC) and Semantic Adherence (SA) using a 0–1 scale. Following the official evaluation protocol (Bansal et al., 2024), we obtain per-model scores by filtering out samples with PC $\leq 0.5$ or SA $\leq 0.5$, and then computing the proportion of samples that pass this joint criterion. For VideoPhy2, which rates PC and SA on a 1–5 scale, we follow the official evaluation practice (Bansal et al., 2025) by filtering out samples with Physics Consistency $\leq 4$ or Semantic Adherence $\leq 4$, and then calculating the proportion of valid samples after filtering. For Qwen2.5-VL (Bai et al., 2025), we follow the evaluation protocol of DreamGen (Jang et al., 2025). Each video is rated on a 0–1 scale using the following prompt:

```
Does the video show good physics dynamics and showcase
a good alignment with the physical world?  Please be
a strict judge.  If it breaks the laws of physics,
please answer 0.  Answer 0 for No or 1 for Yes.  Reply
only 0 or 1.
```

## D.3 INFLUENCING FACTOR EXPERIMENT DETAILS

We conduct two ablation experiments on protocol design factors: context-window length and classifier-free guidance strength. We select five representative scenarios (Ball Drop, Block Slide, Fluid, Cloth and Shadow) from our taxonomy, and five exemplar models: AnimateDiff, ModelScope, CogVideoX, Hunyuan T2V and LTX. For number of frame experiment, we vary the number of input frames in $\{8, 16, 24, 32, 40, 48, 56, 64\}$. AnimateDiff supports up to 32 frames, so we limit its trials accordingly. For CFG experiment, we evaluate multiple classifier-free guidance scales from 1.0 to 8.0 to measure how guidance scale affects the model's ability to discriminate valid from invalid physics cases.

# E APPLICABILITY TO OTHER INTUITIVE PHYSICS BENCHMARKS

We further demonstrate that *LikePhys* applies beyond our main setups by evaluating on the IntPhys Development split (Riochet et al., 2018). The Dev split contains three blocks including O1 (object permanence), O2 (shape constancy), and O3 (spatio-temporal continuity). Each constructed as matched pairs of possible against impossible events. We run *LikePhys* unchanged and report PPE per block and averaged across blocks. As shown in Tab. 6, models exhibit distinct profiles across principles (e.g., some perform better on continuity in O3 than on permanence in O1), mirroring the domain-specific trends we observed on our rigid-body and optics evaluations. This supports the portability of *LikePhys* and its ability to surface principle-level strengths and weaknesses without task-specific tuning.

Table 5: Classification Criterion for Each Physics Law

| Law / Principle | Scenario | Variation key |
|---|---|---|
| *Temporal Continuity* | | |
| | Ball Collision | temporal disorder |
| | Ball Drop | temporal disorder |
| | Block Slide | temporal disorder |
| | Cloth Drape | temporal disorder |
| | Faucet Flow | temporal disorder |
| | Cloth Waving | temporal disorder |
| | Droplet Fall | temporal disorder |
| | Pendulum Oscillation | temporal disorder |
| | Pyramid Impact | temporal disorder |
| | River Flow | temporal disorder |
| | Moving Shadow | temporal disorder |
| | Orbit Shadow | temporal disorder |
| *Spatial Continuity* | | |
| | Ball Collision | teleportation |
| | Ball Drop | teleportation |
| | Block Slide | teleportation |
| | Cloth Waving | flag teleport |
| | Pyramid Impact | teleporting spheres |
| | River Flow | invisible wall |
| | Faucet Flow | teleporting fluid |
| *Conservation of Energy* | | |
| | Ball Drop | over bounce |
| | Ball Collision | momentum amplification |
| | Ball Collision | phantom force |
| | Ball Drop | dynamic scaling |
| | Cloth Waving | elastic explosion |
| | Droplet Fall | self attracting |
| | Pyramid Impact | momentum multiplication |
| | Droplet Fall | non-conservation momentum |
| *Conservation of Mass* | | |
| | Droplet Fall | non-conservation fluid |
| | Droplet Fall | matter creation |
| | Droplet Fall | negative viscosity |
| | Faucet Flow | fracturing fluid |
| | Droplet Fall | phase shifting fluid |
| | River Flow | non-conservation fluid |
| *Geometric Invariance* | | |
| | Ball Collision | size change |
| | Block Slide | size changing |
| | Pyramid Impact | phase shifting |
| | Pyramid Impact | sphere fusion |
| | Orbit Shadow | varying size |
| *Optical Consistency* | | |
| | Moving Shadow | inverted shadow |
| | Moving Shadow | no shadow |
| | Moving Shadow | no object |
| | Moving Shadow | wrong shadow shape |
| | Orbit Shadow | impossible reflection |
| *Material Response* | | |
| | Cloth Drape | impossible folding |
| | Cloth Drape | rubber cloth |
| | Cloth Drape | ground penetration |
| | Block Slide | hovering |
| | Block Slide | irregular motion |
| | Block Slide | jittering |
| | Cloth Waving | flag shatter |

Table 6: **IntPhys Dev (O1/O2/O3) with LikePhys.** We report PPE (%, lower is better) for each block and the average across blocks.

| Model | O3 | O1 | O2 | Avg. |
|---|---|---|---|---|
| Animatediff SDXL | 45.8 | 59.2 | 48.3 | 51.1 |
| Wan2.1-T2V-14B | 48.3 | 51.7 | 52.5 | 50.8 |
| LTX v0.9.5 | 46.7 | 49.2 | 46.7 | 47.5 |
| ModelScope | 40.0 | 52.5 | 49.2 | 47.2 |
| ZeroScope | 42.5 | 48.3 | 50.8 | 47.2 |
| CogVideoX | 43.3 | 49.2 | 46.7 | 46.4 |
| Animatediff | 45.0 | 43.3 | 43.3 | 43.9 |
| CogVideoX-5B | 40.0 | 45.8 | 45.0 | 43.6 |
| Hunyuan T2V | 35.8 | 30.0 | 25.8 | 30.5 |

## F    VISUAL SAMPLES

We further provide visual samples in the constructed benchmark for each scenario as mentioned in Apx. B. As shown in Fig. 7, Fig. 8, and Fig. 9. We show one valid and two invalid cases for each scenario for illustration.

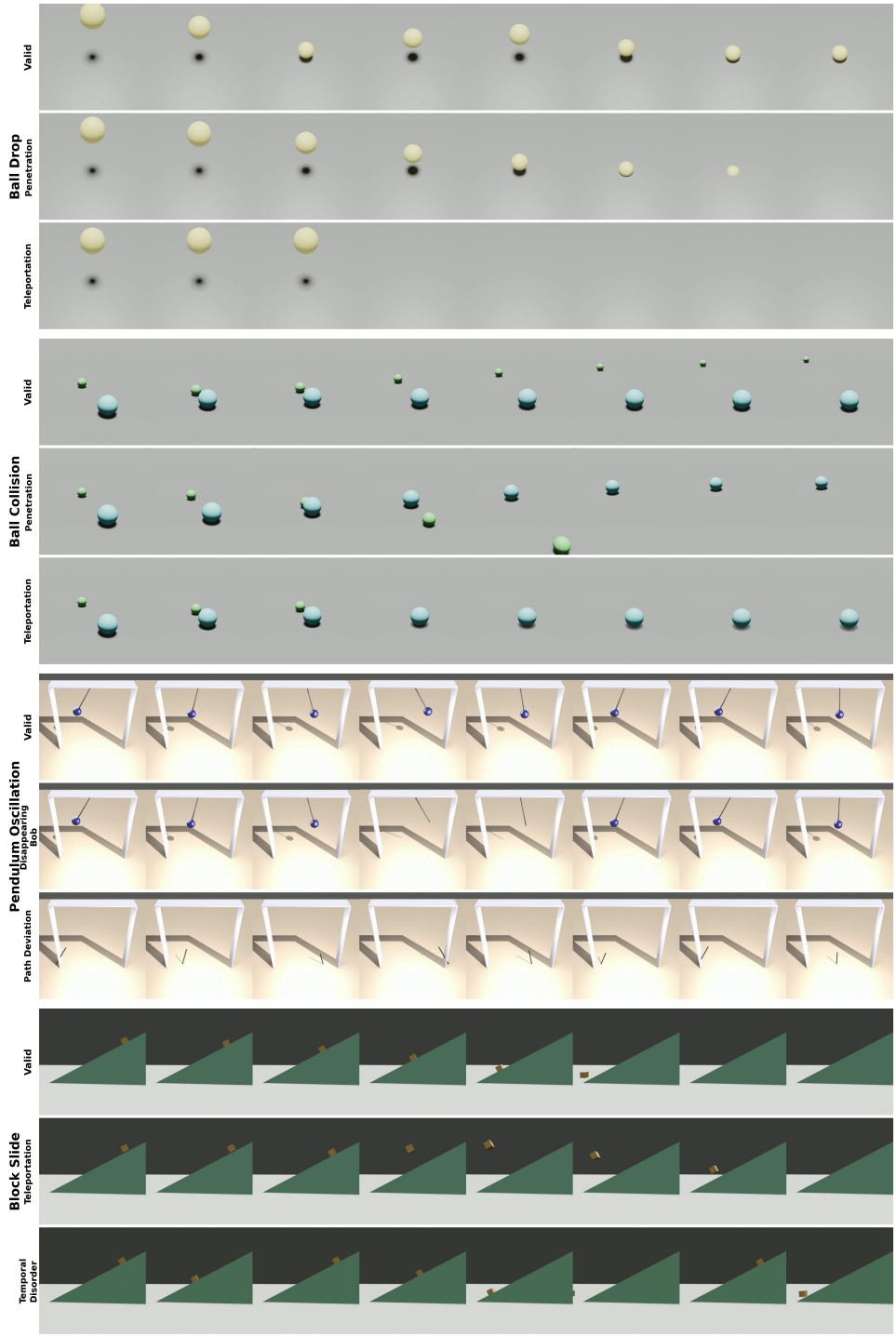

Figure 7: Visual samples from Ball Drop, Ball Collision, Pendulum Oscillation, and Block Slide scenarios.

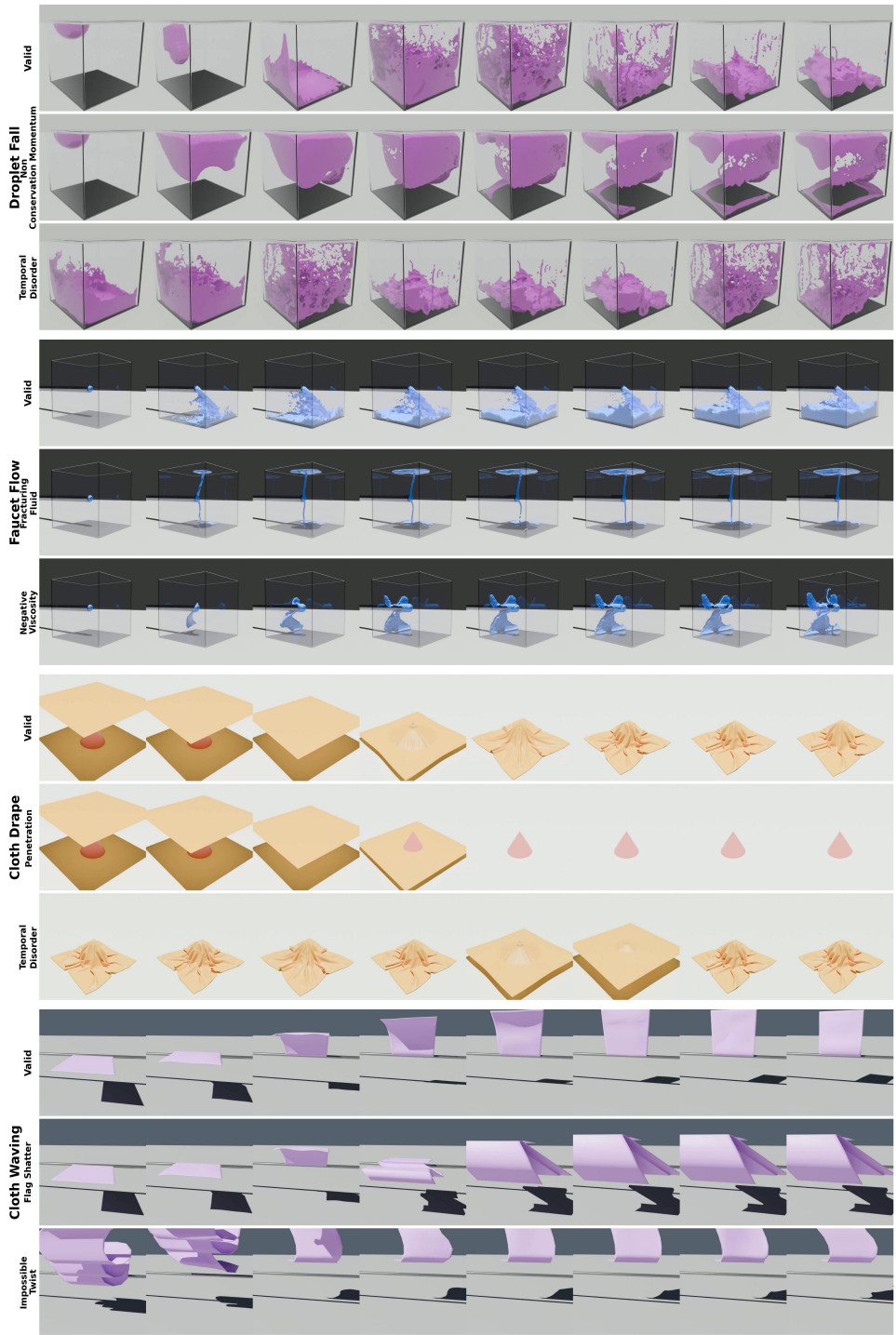

Figure 8: Visual sample from Droplet Fall, Faucet Flow, Cloth Drape, Cloth waving scenarios.

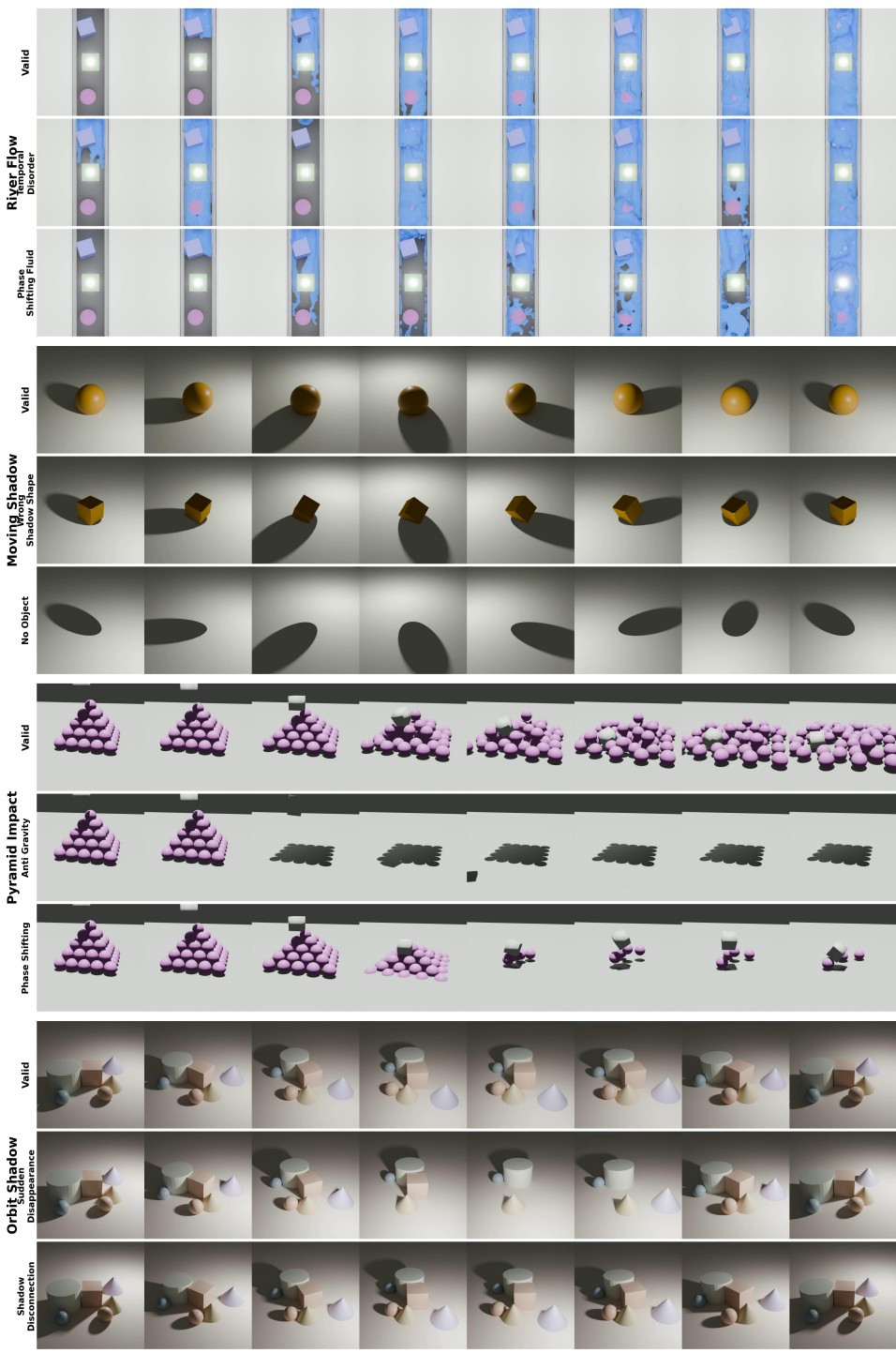

Figure 9: Visual sample from River Flow, Moving Shadow, Pyramid Impact, Orbit Shadow scenario.

