# OpenReview forum: "LikePhys: Evaluating Intuitive Physics Understanding in Video Diffusion Models via Likelihood Preference"
_ICLR.cc/2026/Conference — ICLR 2026 Poster_

### Official Review · Reviewer_y5Dy · 2025-10-26

**Soundness:** 3
**Presentation:** 3
**Contribution:** 3
**Rating:** 6
**Confidence:** 3

**Summary:**

This paper introduces LikePhys, a training-free evaluation framework designed to assess the intuitive physics understanding of video diffusion models (VDMs). Instead of relying on human or vision-language judgments, LikePhys measures how well a model distinguishes physically valid from invalid videos using its denoising loss as a likelihood proxy. The authors construct a controlled benchmark of twelve simulated scenarios across four physics domains—rigid-body, continuum, fluid mechanics, and optical effects—and define the Plausibility Preference Error (PPE) to quantify whether the model assigns higher likelihood to physically plausible sequences.

Through systematic experiments on twelve state-of-the-art VDMs, the study finds that larger Transformer-based models (e.g., Hunyuan T2V, Wan 2.1–14B) outperform UNet-based ones, showing partial emergence of physics reasoning. PPE correlates strongly with human judgments but remains largely independent of visual quality metrics, confirming that it measures physical plausibility rather than appearance. Nonetheless, current models still struggle with complex or chaotic dynamics—particularly in fluid mechanics and conservation-law scenarios—highlighting the need for future work on physics-aware training and longer temporal modeling.

**Strengths:**

1. The paper proposes LikePhys, a novel and training-free method that evaluates intuitive physics understanding through a model’s own likelihood estimation. This approach elegantly connects diffusion models’ denoising objective with physical plausibility assessment, avoiding dependence on human annotation or vision–language model judges. It offers a fundamentally objective and interpretable evaluation paradigm for generative models.
2. LikePhys can be directly applied to any diffusion-based video model without additional training or fine-tuning. By using the denoising loss as an ELBO-based likelihood surrogate, it remains compatible with a wide range of architectures and inference setups, enabling scalable and reproducible benchmarking across different models.
3. The authors construct a highly systematic simulation dataset of twelve physics scenarios covering four domains—rigid-body, continuum, fluid mechanics, and optical effects. Each valid–invalid pair is carefully designed to isolate specific physics violations (e.g., energy, mass, continuity) while holding appearance constant, ensuring the evaluation reflects genuine physics reasoning rather than visual bias.
4. The proposed Plausibility Preference Error (PPE) metric quantifies the proportion of cases where a model fails to prefer physically valid samples. It is intuitive, numerically stable, and easily comparable across models. Moreover, the authors demonstrate that PPE aligns strongly with human judgments of physical consistency while being independent from standard visual quality metrics, confirming its validity and specificity.

**Weaknesses:**

1. A key limitation of the proposed likelihood-preference framework lies in its implicit assumption that higher estimated likelihood (i.e., lower denoising loss) reflects stronger physical understanding. In practice, the likelihood assigned by a diffusion model is influenced by many confounding factors beyond physics correctness. For instance, if a video sample—whether physically valid or invalid—resembles patterns frequently seen in the training data, it may naturally obtain a higher likelihood simply due to data distribution similarity, rather than genuine adherence to physical laws. Conversely, both valid and invalid videos that deviate from the model’s training distribution could be assigned uniformly low likelihoods, making the difference between them statistically insignificant. As a result, the LikePhys metric may sometimes conflate distribution familiarity with physical plausibility, leading to inaccurate or unstable evaluations, especially when the model exhibits strong dataset bias. This issue raises questions about the robustness and interpretability of using likelihood differences alone as a proxy for intuitive physics understanding.
2. Although the benchmark covers four major physics domains, it remains synthetic and controlled, relying solely on Blender-rendered simulations. While this ensures experimental rigor, it limits the method’s ability to generalize to real-world, noisy, or unstructured videos, where visual complexity, uncertainty, and imperfect physical consistency are common. The results might therefore overestimate models’ real-world physics reasoning ability.
3. The metric treats intuitive physics understanding as a pairwise likelihood preference problem, which captures surface-level plausibility but may fail to reflect causal reasoning, temporal prediction, or long-horizon dynamics that are essential for deeper physical understanding. As a result, models that memorize motion patterns could perform well on LikePhys without genuinely learning the underlying physical principles.
4. While the paper reports domain-level PPE scores, it provides limited qualitative analysis or visual diagnosis of why certain models fail under specific laws (e.g., temporal continuity or conservation of mass). More detailed case studies or ablation examples could have strengthened interpretability and clarified whether errors stem from architecture limitations, data bias, or diffusion noise modeling.

**Questions:**

I do not see evaluation code and data in the supplementary material. Do you have any opensource plan?

---

> ### Author Response · Authors · 2025-11-20
>
> Dear Reviewer y5Dy,
>
>
> Thank you for your positive and insightful feedback and suggestions. We appreciate your recognition of the novelty of LikePhys as a training-free, likelihood-based evaluation that links diffusion denoising to physics plausibility, its broad applicability enabling scalable and reproducible benchmarking across diverse architectures and inference settings, and the PPE metric’s intuitiveness, stability, and strong alignment with human judgments while remaining independent of standard visual-quality metrics. We also thank you for raising the insightful and interesting question about the difference between pattern recognition and genuine physics understanding, which we are happy to discuss further. We address your questions below.
>
>
>
>
> **Q1: A key limitation of the proposed likelihood-preference framework lies in its implicit assumption that higher estimated likelihood (i.e., lower denoising loss) reflects stronger physical understanding... This issue raises questions about the robustness and interpretability of using likelihood differences alone as a proxy for intuitive physics understanding.**
>
>
>
>
> **A1**: Thanks for raising this interesting point.
> Regarding the distinction between “pattern memorization” and “genuine physics understanding”, we think this is still an open conceptual question in general.
> Our method is concerned with whether the distribution learned by a video generative model is close to a physics-plausible distribution, by comparing the likelihoods the model assigns to physically valid and invalid video pairs under controlled conditions. In this work, we take an operational view: we explicitly define a model’s intuitive physics capacity as its ability to assign higher likelihood to physically plausible videos than to invalid ones in these controlled comparisons. Several factors can influence this capacity, such as the training data distribution, suboptimal optimization, and the training objective, but we do not make any prior assumptions about which of these is responsible.
>
>
> About your concern that likelihood may reflect “distribution familiarity” rather than “genuine physics understanding”: in our setup, these two are not independent. If a model has been trained on data that frequently contains physically implausible patterns, then this is precisely a form of poor physics understanding we want to detect. As an intuitive thought experiment, consider a model trained purely on physically impossible videos (strong dataset bias), where object size constantly changes and energy spontaneously increases. If the model learns this distribution well, it will assign high likelihood to such impossible videos and low likelihood to physically correct ones. Under our evaluation, this model would obtain a high PPE, because it systematically prefers invalid over valid videos. At the same time, if we sample from this model, its generated videos would be likely judged as physically implausible. In this case, the “low physics understanding” is caused by dataset bias, but this is exactly what PPE is meant to capture.
>
>
> The second part of the concern is the case where both valid and invalid videos are far from the training distribution and receive uniformly low likelihood. Here, our protocol uses pairwise comparisons rather than raw likelihoods. Within each variation, valid and invalid videos share the same prompt, camera, lighting, textures, and geometry, and differ only by a controlled physics violation. If the model truly cannot distinguish them at all, then the likelihood difference will fluctuate around zero and PPE will approach 0.5, which we interpret as the model not expressing a clear physics preference on that scenario. This is a meaningful outcome: it indicates that, for this benchmark, the model does not exhibit strong intuitive physics (i.e. more than chance level). In practice, we also average PPE over many pairs and scenarios, which reduces the impact of individual noisy comparisons and yields a stable estimate (as also supported by our empirical results).
>
>
> Overall, our method does not claim that likelihood itself is a “pure” physics signal. Instead, it uses likelihood differences in carefully controlled valid–invalid pairs to probe how closely the learned model distribution aligns with physics-plausible behavior. Dataset bias and training distribution do influence this alignment, but they are part of what LikePhys is designed to measure rather than something we attempt to factor out. Finally, we have also empirically verified the correlation between PPE and downstream video generation physics plausibility in the human study, which supports the robustness and interpretability of PPE as a proxy for intuitive physics understanding.

---

> > ### Author Response · Authors · 2025-11-20
> >
> > **Q2: Although the benchmark covers four major physics domains, it remains synthetic and controlled, relying solely on Blender-rendered simulations. While this ensures experimental rigor, it limits the method’s ability to generalize to real-world, noisy, or unstructured videos, where visual complexity, uncertainty, and imperfect physical consistency are common. The results might therefore overestimate models’ real-world physics reasoning ability.**
> >
> >
> >
> >
> > **A2**: Our method follows the classic violation-of-expectation paradigm [1,2], which is mostly built on simulated data and has been successful in evaluating the physics understanding of various vision models. It is essentially impossible to obtain large-scale real-world test data with precisely controlled violations of physical laws, so simulation is a natural choice for isolating physics.
> > While real-world videos are indeed more visually complex and often contain imperfect physics, this does not exempt models from making errors on much simpler, controlled violations. In our benchmark, each valid–invalid video pair differs only by a specific, well-defined physics violation. This allows us to attribute failures to particular laws or scenarios and provides a lower bound on a model’s physics capacity: if a model struggles on these controlled cases, it is unlikely to handle more noisy, unstructured real-world scenes.
> >
> >
> > In the evaluation protocol, we deliberately control confounding factors through simulation. Each physics scenario is designed to be relatively simple, with clearly attributable governing dynamics and consistent appearance. Within a variation, valid and invalid videos share the same prompt, camera, lighting, textures, and geometry, and differ only in physics adherence. Our method then compares the relative likelihood within each valid–invalid pair, rather than the absolute likelihood of a single sample. This pairwise design is intended to cancel out model-specific responses to visual appearance or style, so that the remaining likelihood difference arises from the controlled physics violation. Overall, the synthetic setting is used to isolate and probe physics in a controlled way; it does not claim to capture the full complexity of real-world videos, but it provides a rigorous and interpretable test of whether models respect basic physical regularities under matched visual conditions.
> >
> >
> > [1] Riochet, R., Castro, M.Y., Bernard, M., Lerer, A., Fergus, R., Izard, V. and Dupoux, E., 2018. Intphys: A framework and benchmark for visual intuitive physics reasoning. arXiv preprint arXiv:1803.07616.
> >
> >
> > [2] Bordes, F., Garrido, Q., Kao, J.T., Williams, A., Rabbat, M. and Dupoux, E., 2025. IntPhys 2: Benchmarking Intuitive Physics Understanding In Complex Synthetic Environments. arXiv preprint arXiv:2506.09849.
> >
> >
> >
> >
> >
> >
> >
> >
> > **Q3: The metric treats intuitive physics understanding as a pairwise likelihood preference problem, which captures surface-level plausibility but may fail to reflect causal reasoning, temporal prediction, or long-horizon dynamics that are essential for deeper physical understanding. As a result, models that memorize motion patterns could perform well on LikePhys without genuinely learning the underlying physical principles.**
> >
> >
> >
> >
> > **A3**: Thanks for raising this interesting point.
> > As discussed in our response to Q1, we take an operational view and define a model’s intuitive physics capacity as its ability to assign higher likelihood to physically valid videos than to invalid ones under controlled violations. LikePhys is therefore designed to test whether the learned distribution aligns with physics-plausible patterns, irrespective of whether the internal mechanism is “pattern memorization” or an explicit causal model.
> > Regarding the concern about “surface-level plausibility”, several of our scenarios already require non-trivial temporal reasoning (e.g., tracking momentum exchange over time, delayed collisions, occlusions), so a model cannot succeed purely from a single static frame. As shown in our main results (Sec. 4.1 and Sec. 4.5), current video diffusion models still make systematic errors on these relatively simple controlled valid–invalid pairs, which suggests that their physics capacity is limited even before considering more complex, long-horizon settings.
> > We agree that PPE does not cover all aspects of deep causal reasoning or arbitrarily long-term prediction, and we do not claim it is a complete test of physics understanding. It is a first-step, controlled pairwise probe under matched visual conditions. Extending this framework to longer horizons, multi-step counterfactuals, or explicitly causal tasks is a natural direction for future work. Importantly, our human study shows that PPE already correlates well with human judgments of physical plausibility in downstream text-to-video generation, indicating that it captures a practically relevant component of “physics quality” observed by users.

---

> > > ### Author Response · Authors · 2025-11-20
> > >
> > > **Q4: While the paper reports domain-level PPE scores, it provides limited qualitative analysis or visual diagnosis of why certain models fail under specific laws (e.g., temporal continuity or conservation of mass). More detailed case studies or ablation examples could have strengthened interpretability and clarified whether errors stem from architecture limitations, data bias, or diffusion noise modeling.**
> > >
> > >
> > >
> > >
> > > **A4**:In Section 4.3 and Figure 5 we also provide a finer-grained analysis at the level of individual physics laws. Using the mapping defined in Appendix A (Table 5), we group failure modes into seven law categories and report average PPE per law. This analysis shows, for example, that temporal continuity exhibits the largest variance across models, pointing to unstable long-range coherence when motion spans many frames or involves occlusion. Spatial continuity and conservation laws (e.g., momentum and mass) also have relatively high errors, which is consistent with the lack of explicit global constraints in standard training objectives and samplers. In contrast, geometric invariance and optical consistency tend to be better satisfied, likely reflecting priors inherited from large-scale static image and short-clip training. Material response falls in between, indicating that contact events, friction and surface compliance remain challenging. In general, these patterns act as a diagnostic view of which types of laws are most problematic and suggest that many errors are linked to limited temporal context and missing global constraints rather than a single architectural detail. We now clarify this interpretation in Section 4.3 and explicitly point readers to the law-mapping rules in Appendix A, which make the connection between invalid patterns and each physics law transparent.

---

> ### Comment · Reviewer_y5Dy · 2025-11-24
>
> The authors have addressed most of my concerns. I keep my score to support the paper.

---

### Official Review · Reviewer_tiXF · 2025-10-29

**Soundness:** 2
**Presentation:** 2
**Contribution:** 2
**Rating:** 2
**Confidence:** 5

**Summary:**

The problem studied in this paper is how to evaluate a video generation model w.r.t. obedience to physical rules, under the setting that the model weights are accessible. The proposed method is train-free/zero-shot: 1) synthesize benchmark videos with Blender with two sets, $V+$ for physical feasible videos and $V-$ for physical infeasible videos; 2) calculate noise estimation errors for videos in $V+$ and $V-$; 3) calculate how many times the video model would estimate lower errors for videos from $V+$ videos from $V-$, then normalize the score to get the so called "Plausibility Preference Error" score (page 4, line 211~213).
With the proposed method, authors benchmarked 12 open-sourced video generation models, and compared its correlation with human as well as VLM judgements.

**Strengths:**

Although the work is still crude in its current form, it provides an insightful view angle for video model evaluation, with its train-free method.

**Weaknesses:**

The idea of evaluating an image/video generation model's certain capability with its noise estimation error is not new, though this is the first work I met for physical video generation.  Plus there's some essential problems with the proposed method unclear from the paper, which is stated in the following "Questions" section.

**Questions:**

1. About the mismatch between the assumption and the actual calculation of PPE: in page 1, line 52~53, authors stated the essential assumpation *we use a simulator to render paired videos. In one, we render physically-realistic phenomena, while in the other we introduce
a controlled violation of physics. We keep the visual appearance consistent in the pair, ensuring that any difference resulting from processing the videos can be attributed solely to the breach of physics principles*, which means the video $x^+$ and $x^-$ should have the same prompt and scene content. But from the eq.5 in page 4 and appendix A in page 14, it seems non-paired videos $x^+_j$ and $x^-_k$ are also included in calculation. Did I get anything wrong or the presented calculation is problematic here?
2. Distribution discrepancy among the benchmark videos and tested models: from the demo frames in Fig.1 and appendix page 20~22, it seems the synthesized video from Blender are over simplified, with non-realistic looking and blank background. This distribution would be definitely far from the trainset of video generation models tested in the paper. Though results in Table 2 show that the method proposed is better than Qwen2.5 VL and VideoPhy1/2, I doubt whether this is fair for models with less training data similar to the synthesized video from the paper. From Table 1 in page 6, it seems the calculated PPE scores fluctuate dramatically among the tested 12 physics categories for any of the 12 models listed there, and I doubt the distribution discrepancy might be the hidden problem.
3. Following question 2, authors didn't specify which one from the Qwen2.5 VL model family is used.

---

> ### Author Response · Authors · 2025-11-20
>
> Dear reviewer tiXF,
>
>
> Thank you for your constructive feedback and valuable suggestions. We appreciate that you recognize our work provides an insightful new perspective on video model evaluation through a training-free approach. We have now polished the writing according to the suggestions and feedback in the updated manuscript.  We now address your concerns below.
>
>
>
>
> **Q1: The idea of evaluating an image/video generation model's certain capability with its noise estimation error is not new, though this is the first work I met for physical video generation.**
>
>
> **A1**: The violation-of-expectation paradigm is a classic method to evaluate the intuitive physics of vision models. It has been shown to be successful across various vision models, yet its application in video generative models is not well-explored due to the unclear approach to formulating a generative model for a discriminative task. We consider extending this promising paradigm to video generation to be non-trivial and consider this as a meaningful research gap and thus deliver this work. If you had specific prior work in mind that uses noise estimation error in a similar way for physics evaluation in video generation, we would greatly appreciate the references. We are happy to further clarify the distinction and properly acknowledge them in the revised manuscript.
>
>
>
>
>
>
>
>
>
>
>
>
> **Q2:About the mismatch between the assumption and the actual calculation of PPE … Did I get anything wrong or the presented calculation is problematic here?**
>
>
>
>
> **A2**: Thank you for raising this point. Our PPE calculation does not involve non-paired videos. Specifically, for each physics scenario, we create 10 controlled variations. For each variation, we construct two sets of videos following the violation-of-expectation paradigm: a valid set with M samples and an invalid set with N samples. Within a given variation, all videos (M valid and N invalid) share the same visual style and are simple and consistent in visual appearance; the only differences arise from controlled violations of the governing physics dynamics. We then calculate a variation-wise PPE using N*M pairwise comparisons as described in Eq. (5) and aggregate over all variations to quantify the PPE for each physics scenario.
> We have now revised Sections 3.2 and 3.3 (Methodology) to explicitly clarify these and remove any potential ambiguity and confusion.

---

> > ### Author Response · Authors · 2025-11-20
> >
> > **Q3: Distribution discrepancy among the benchmark videos and tested models: from the demo frames in Fig.1 and appendix page 20~22, it seems the synthesized video from Blender are over simplified, with non-realistic looking and blank background. This distribution would be definitely far from the trainset of video generation models tested in the paper. Though results in Table 2 show that the method proposed is better than Qwen2.5 VL and VideoPhy1/2, I doubt whether this is fair for models with less training data similar to the synthesized video from the paper. From Table 1 in page 6, it seems the calculated PPE scores fluctuate dramatically among the tested 12 physics categories for any of the 12 models listed there, and I doubt the distribution discrepancy might be the hidden problem.**
> >
> >
> > **A3**: First, regarding the simulator-generated data, we design each physics scenario to be relatively simple, with clearly attributable governing physics dynamics and consistent appearance. While the Blender videos are indeed stylized and far from the training distribution of current video generators, our method is constructed to be insensitive to such style differences. We do not use the absolute likelihood of a single video, but the likelihood difference between a valid–invalid pair that shares the same prompt and scene content. By comparing relative likelihood within a controlled pair, model-specific responses to visual style and other confounding factors cancel out, and the remaining difference is driven by the controlled physics violation. In this way, model bias toward particular visual styles should not dominate the PPE metric, and the metric remains comparable across models even when the benchmark distribution is simplified. This follows the classic violation-of-expectation paradigm for evaluating physics understanding in vision models, where there is also a distribution shift between the models’ training data and the simulator-generated test data [1,2,3]. Prior work has shown that this setting can still yield meaningful and robust conclusions, which motivates us to adopt the same practice.
> >
> >
> > Second, regarding the human study and comparison to other evaluators, we do not use the Blender synthetic data for human annotation. Instead, we use standard text-to-video generation settings to generate videos with various video generative models and then collect human ratings on these generated videos. We follow the standard paradigm [4] for VLM-based video physics plausibility assessment: given text prompts, models generate videos, and human annotators rate their physical plausibility. We then study how well PPE, computed on our synthetic benchmark, correlates with human judgments on these downstream generations, and we find that PPE aligns well with human preference.
> > We have revised Section 4 to better highlight the procedure for the human study, and Sections 1 (Introduction) and 3 (Methodology) to clarify these points and remove potential ambiguity.
> >
> >
> > [1] Riochet, R., Castro, M.Y., Bernard, M., Lerer, A., Fergus, R., Izard, V. and Dupoux, E., 2018. Intphys: A framework and benchmark for visual intuitive physics reasoning. arXiv preprint arXiv:1803.07616.
> >
> >
> > [2] Bordes, F., Garrido, Q., Kao, J.T., Williams, A., Rabbat, M. and Dupoux, E., 2025. IntPhys 2: Benchmarking Intuitive Physics Understanding In Complex Synthetic Environments. arXiv preprint arXiv:2506.09849.
> >
> >
> > [3] Garrido, Q., Ballas, N., Assran, M., Bardes, A., Najman, L., Rabbat, M., Dupoux, E. and LeCun, Y., 2025. Intuitive physics understanding emerges from self-supervised pretraining on natural videos. arXiv preprint arXiv:2502.11831.
> >
> >
> > [4] Bansal, H., Peng, C., Bitton, Y., Goldenberg, R., Grover, A. and Chang, K.W., 2025. Videophy-2: A challenging action-centric physical commonsense evaluation in video generation. arXiv preprint arXiv:2503.06800.
> >
> >
> >
> >
> >
> >
> > **Q4: Following question 2, authors didn't specify which one from the Qwen2.5 VL model family is used.**
> >
> >
> > **A4**: We use Qwen2.5 VL 7B-Instruct, following the same practice as DreamGen [5]. We have revised Section 4 to specify this detail.
> >
> >
> > [5] Jang, J., Ye, S., Lin, Z., Xiang, J., Bjorck, J., Fang, Y., Hu, F., Huang, S., Kundalia, K., Lin, Y.C. and Magne, L., 2025. DreamGen: Unlocking Generalization in Robot Learning through Video World Models. arXiv preprint arXiv:2505.12705.

---

### Official Review · Reviewer_Bi5p · 2025-11-03

**Soundness:** 3
**Presentation:** 4
**Contribution:** 1
**Rating:** 4
**Confidence:** 4

**Summary:**

This paper proposes a very simple approach to eval video diffusion models' physical plausiblity, named Plausibility Preference Error (PPE). PPE is a pair-wise eval metrics, by calculating the error between the loss of denoising physically valid and impossible videos.

**Strengths:**

1. This eval method is easy to understand and implement, the motivation is clear.
2. The authors provide good analysis on Disentanglement of Visual Appearance, convinced me that PPE is mainly for physical plausiblity, and are merely affected by visual quality.
3. The authors provide good discussions for model size, number of frames, CFG, data scale, which provides useful insights.

**Weaknesses:**

1. This approach needs to access model params, which makes it impossible for evaluating closed-source video diffusion models. This is also discussed in Section 5
2. All of the scenarios are from simulator, will the models pretrained more on simulated data have advantages over other models? I see some discussion in Section 5, the authors rely on an assumption that the models are mostly trained on real-world recordings rather than
animated or synthetic content. But I doubt this may not hold.
3. One of the most important motivation of this paper is "VLM based methods fail to disentangle physics from visual appearance.", while the authors do not provide a correlation between these scores and visual quality metrics like In Table 3. I felt this is important to justify the necessity of using PPE

**Questions:**

1. For timestep selection, different models are using different weighting scheme, having differnt density of timestep sampling when doing training, so for the timestep selection, does it makes sense to sample uniformly? or randomly sampled? Is such sampling strategy robust enough?

---

> ### Author Response · Authors · 2025-11-20
>
> Dear Reviewer Bi5p,
>
>
> Thank you very much for your insightful review and constructive comments. We appreciate that you find our method easy to understand and implement, with clear motivation. We also appreciate that you value our analysis on disentanglement of visual appearance, as well as discussions regarding model size, number of frames, CFG, and data scale which provide useful insights. We now address your concerns below.
>
>
>
>
> **Q1: This approach needs to access model params, which makes it impossible for evaluating closed-source video diffusion models. This is also discussed in Section 5**
>
>
> **A1**:  Thank you for raising this point. Our method indeed requires access to the denoising network, which makes it most naturally applicable to open-source or internally accessible models. We argue that this requirement is less restrictive in the open-source community, where open-source video diffusion models play an important role and are widely used as research baselines.
> For closed-source models, our method is still useful from the provider’s side: it can be integrated into the internal evaluation pipeline to monitor training progress, diagnose physics-related failure modes, and select checkpoints for release to users.
> In general, we see our approach as a novel contribution.  To our knowledge, the first method that evaluates physics in diffusion models directly through the denoising process, without requiring video generation, and can therefore serve as a valuable tool for both open-source and proprietary model development.
>
>
>
>
> **Q2: All of the scenarios are from simulator, will the models pretrained more on simulated data have advantages over other models? I see some discussion in Section 5, the authors rely on an assumption that the models are mostly trained on real-world recordings rather than animated or synthetic content. But I doubt this may not hold.**
>
>
> **A2**: Our method follows the classic violation-of-expectation paradigm [1,2], which is mostly built on simulated data and has been successful in evaluating the physics understanding of various vision models. It is essentially infeasible to obtain test data from the real world that contains controlled violations of physical laws at scale, so simulation is a natural and widely adopted choice.
> Regarding whether a model trained more on synthetic data would have an advantage, our method is designed to provide an unbiased estimation of physics plausibility preference for the following reason: we do not measure the absolute likelihood of a single video, but rather the likelihood difference within a valid–invalid pair. As long as the two videos in a pair share the same visual appearance and differ only in physics adherence, the pairwise comparison effectively cancels out the influence of visual style and other appearance factors. Thus, the difference in estimated likelihood remains meaningful, can be attributed to physics violations, and yields a metric that is comparable across models, regardless of whether they were trained more on realistic or synthetic data.
>
>
>
>
> Regarding the assumption that models are mostly trained on real-world recordings, our method does not rely on this assumption. Our method aims to assess whether the learned video generative model distribution is close to a physics-plausible distribution by comparing the likelihoods the model assigns to physically valid and invalid video pairs. The training data distribution is one factor that may influence this physics capacity, but we do not make any prior assumptions about it. We have revised the discussion in Section 5 in the updated manuscript to remove potential ambiguity on this point.
>
>
>
>
>
>
>
>
> [1] Riochet, R., Castro, M.Y., Bernard, M., Lerer, A., Fergus, R., Izard, V. and Dupoux, E., 2018. Intphys: A framework and benchmark for visual intuitive physics reasoning. arXiv preprint arXiv:1803.07616.
>
>
> [2] Bordes, F., Garrido, Q., Kao, J.T., Williams, A., Rabbat, M. and Dupoux, E., 2025. IntPhys 2: Benchmarking Intuitive Physics Understanding In Complex Synthetic Environments. arXiv preprint arXiv:2506.09849.

---

> > ### Author Response · Authors · 2025-11-20
> >
> > **Q3: One of the most important motivation of this paper is "VLM based methods fail to disentangle physics from visual appearance.", while the authors do not provide a correlation between these scores and visual quality metrics like In Table 3. I felt this is important to justify the necessity of using PPE**
> >
> >
> > **A3**: For the VLM-based method, numerous works have found that VLM-as-judge have various limitations in subject bias [3] and interpretative variance [4]. Specifically regarding assessing physics evaluation, the literature has shown limited physical reasoning capacity in such approach [5]. On the other hand violation-of-expectation paradigm has been shown to be successful on various vision models[1,2], yet its application in video generative models is not well-explored due to the unclear way to form a generative model to perform discriminative task. We identify this as a meaningful research gap and thus deliver this work.
> >
> >
> >
> >
> > We also conduct the additional correlation analysis to show the difficulty in disentangling visual appearance in video physics evaluation. As shown in the table below:
> >
> >
> >
> >
> > | Metric                 | Corr. w. PPE | Corr. w. VideoPhy2 | Corr. w. VideoPhy1|
> > |:-----------------------|-------------:|--------------------------------:|-------------------------------------------:|
> > | Subject Consistency    |        **-0.01** |                            0.19 |                                       0.35 |
> > | Background Consistency |       **-0.01** |                            0.28 |                                       0.34 |
> > | Aesthetic Quality      |       **-0.05** |                           -0.23 |                                      -0.08 |
> > | Motion Smoothness      |         0.15 |                            **0.08** |                                       0.14 |
> > | Temporal Flickering    |        **0.12** |                            **0.12** |                                       0.17 |
> >
> >
> >
> >
> > Moreover, we have shown in Tab 2 on the human study that PPE can serve as a strong proxy for downstream-generated video physics plausibility. In the human study, we follow the standard VLM-based video physics plausibility evaluation paradigm by generating a set of videos with a text prompt and annotating human preference to calculate correlation with the automatic evaluation metric. We believe this can serve as evidence for the practical significance of PPE.
> >
> >
> > [3]Wu, M. and Aji, A.F., 2025, January. Style over substance: Evaluation biases for large language models. In Proceedings of the 31st International Conference on Computational Linguistics (pp. 297-312).
> >
> >
> > [4]Li, D., Jiang, B., Huang, L., Beigi, A., Zhao, C., Tan, Z., Bhattacharjee, A., Jiang, Y., Chen, C., Wu, T. and Shu, K., 2025, November. From generation to judgment: Opportunities and challenges of llm-as-a-judge. In Proceedings of the 2025 Conference on Empirical Methods in Natural Language Processing (pp. 2757-2791).
> >
> >
> > [5]Motamed, S., Chen, M., Van Gool, L. and Laina, I., 2025. TRAVL: A Recipe for Making Video-Language Models Better Judges of Physics Implausibility. arXiv preprint arXiv:2510.07550.
> >
> >
> >
> >
> >
> >
> >
> >
> > **Q4: For timestep selection, different models are using different weighting scheme, having differnt density of timestep sampling when doing training, so for the timestep selection, does it makes sense to sample uniformly? or randomly sampled? Is such sampling strategy robust enough?**
> >
> >
> >
> >
> > **A4**: We presented additional experiment in Appendix E (Fig 6) regarding timestep selection. Specifically, we show that (1) the overall valid–invalid preference trend is consistent across most timesteps for the tested models, and (2) the timestep at which the valid–invalid separation peaks varies across models and scenarios. These observations suggest that oversampling any particular region of the diffusion trajectory could bias the estimate toward that region. Motivated by this, we adopt a uniform sampling over timesteps when computing PPE, which avoids privileging any specific noise level. This also aligns with prior work that uses diffusion models as density estimators [6], where integrating over the whole trajectory rather than focusing on a narrow band is important for robustness.
> >
> >
> >
> >
> >
> >
> >
> >
> > [6] Li, A.C., Prabhudesai, M., Duggal, S., Brown, E. and Pathak, D., 2023. Your diffusion model is secretly a zero-shot classifier. In Proceedings of the IEEE/CVF International Conference on Computer Vision (pp. 2206-2217).

---

### Official Review · Reviewer_hTWK · 2025-11-03

**Soundness:** 3
**Presentation:** 3
**Contribution:** 2
**Rating:** 4
**Confidence:** 4

**Summary:**

This paper introduces LikePhys, a training-free, likelihood-preference-based evaluation method designed to assess the intuitive physics understanding of video diffusion models (VDMs). The core motivation is to address the challenge of disentangling physical plausibility from visual appearance in generated videos—an issue that existing evaluation methods often fail to handle due to biases from visual fidelity or subjective judgments. The key contribution of this work is the development of a new evaluation framework along with a comprehensive benchmark.

**Strengths:**

- This work is the first to utilize diffusion model likelihoods for evaluating intuitive physics. The approach of using the denoising objective as a likelihood surrogate for a "violation-of-expectation" test is both clever and well-justified, as it effectively examines the model's internal representation of physical concepts.

- Experimental results indicate that the proposed Physics Perceptual Evaluation (PPE) aligns more closely with human preferences compared to existing automatic metrics.

- The benchmark is comprehensive, providing a valuable framework for evaluating these models.

**Weaknesses:**

- The primary concern is that the method's validity depends on a curated set of synthetic simulations. While controlled violations are necessary, this raises questions about how well the findings generalize to the distribution of real-world, natural videos.

- As stated in the paper, this work assumes the learned distribution is physics-plausible, which is actually infeasible for large video diffusion models.

**Questions:**

- The benchmarks were simulated using Blender. Which physics engine did you use, and what is the duration of each video?

- In Figure 3, the textures appear relatively simple. Have you included more complex textured objects?

- I feel that this approach is somewhat similar to Direct Preference Optimization (DPO). Could the author please elaborate on this point?

- If the model not only trained on realistic videos, how to adapt your work to evaluate this model?

---

> ### Author Response · Authors · 2025-11-20
>
> Dear Reviewer hTWK,
>
> Thank you very much for your insightful reviews and constructive comments. We appreciate that you recognize our work as the first to leverage diffusion model likelihoods for evaluating intuitive physics, and that you find our approach both clever and well-justified, with a comprehensive benchmark that provides a valuable framework. We now address your concerns below.
>
> **Q1: The primary concern is that the method's validity depends on a curated set of synthetic simulations. While controlled violations are necessary, this raises questions about how well the findings generalize to the distribution of real-world, natural videos.**
>
> **A1**: Thank you for raising this point. Our approach follows the violation-of-expectation paradigm [1,2], which is mostly built on simulated data and has been successful in evaluating the physics understanding of various vision models. As you noted, it is essentially infeasible to obtain real-world videos with controlled physics violations. In our benchmark, we therefore use simulation to isolate the governing physics: within each preference pair, we deliberately match visual appearance (camera, lighting, textures, composition) so that valid and invalid videos differ only in physical plausibility. This design ensures that any likelihood difference within a pair can be attributed to violations of the underlying physics law rather than to confounding visual factors.
> Regarding generalization to real-world, natural videos, our human study directly targets this question. We generate text-to-video samples with various video diffusion models and ask human annotators to rate their physical plausibility. We then compare these human-based rankings to the rankings induced by PPE. We find that PPE serves as a good proxy for human preference on downstream text-to-video generation, indicating that the physics capacity measured on our synthetic benchmark meaningfully transfers to realistic use cases of video generative models in downstream generation.
>
> [1] Riochet, R., Castro, M.Y., Bernard, M., Lerer, A., Fergus, R., Izard, V. and Dupoux, E., 2018. Intphys: A framework and benchmark for visual intuitive physics reasoning. arXiv preprint arXiv:1803.07616.
>
> [2] Bordes, F., Garrido, Q., Kao, J.T., Williams, A., Rabbat, M. and Dupoux, E., 2025. IntPhys 2: Benchmarking Intuitive Physics Understanding In Complex Synthetic Environments. arXiv preprint arXiv:2506.09849.
>
> **Q2: As stated in the paper, this work assumes the learned distribution is physics-plausible, which is actually infeasible for large video diffusion models.**
>
> **A2**: Our method does not rely on the assumption that the video diffusion models are trained on realistic videos. Measuring if the learned model distribution is aligned with a real-world physics-plausible distribution is the aim of our method, rather than its assumption. Our method measures the video diffusion model’s physics understanding defined from a distribution perspective, where we use the estimated likelihood difference over controlled invalid-valid video pairs as a proxy. The idea is that if a video diffusion model can assign a low likelihood to an invalid video (i.e. low probability on a physically plausible distribution) and vice versa, we consider the learned model distribution to be close to the physically plausible distribution, and thus have a good physics understanding.
>
>
> We have revised Section 5 of the discussion to address the ambiguity and misunderstanding, better highlighting our motivations and method assumptions.
>
>
>
> **Q3: The benchmarks were simulated using Blender. Which physics engine did you use, and what is the duration of each video?**
>
> **A3**: All twelve scenarios are implemented in Blender 4.4.3 using its native physics engines and standard renderers. For rigid-body dynamics (Ball Drop, Ball Collision, Block Slide, Pyramid Impact), we use Blender’s Bullet rigid-body engine via the scene rigid body world; Ball Drop and Ball Collision are rendered with Cycles, while Block Slide and Pyramid Impact are rendered with Eevee Next. For deformable solids, Cloth Drape and Cloth Waving use Blender’s Cloth solver and are rendered with Cycles, and the gelatine/deformable drop scenario uses the Soft Body solver, also rendered with Cycles. For fluid scenarios (Faucet Flow, Droplet Fall, River Flow), we use Mantaflow in liquid mode as the physics engine and render all three with Cycles. For the Moving/Orbit Shadow scenarios, there is no rigid/cloth/fluid/soft-body solver active; shadows are produced purely by Cycles ray-traced lighting and animated geometry. For video duration, we standardize all clips to consist of 60 rendered frames in 24 fps.

---

> > ### Author Response · Authors · 2025-11-20
> >
> > **Q4 In Figure 3, the textures appear relatively simple. Have you included more complex textured objects?**
> >
> > **A4**: We include additional qualitative examples in the appendix. Our goal, however, is not to maximize visual complexity, but to maintain consistency in visual appearance within each controlled variation so that valid and invalid videos differ only in the underlying physical dynamics. Even if a model prefers a particular visual style or texture, that preference is shared by both videos in a pair and thus cancels out in the likelihood preference calculation. In this way, our protocol disentangles visual appearance from physics understanding.
> >
> >
> > **Q5: I feel that this approach is somewhat similar to Direct Preference Optimization (DPO). Could the author please elaborate on this point?**
> >
> > **A5**: Thanks for rasing this interesting point. Our method and DPO are both preference-based, but they operate at different levels and serve different purposes. DPO is a training framework that updates a generative model using human (or synthetic) pairwise preferences, typically by encouraging higher log-probability on preferred samples relative to rejected ones. In contrast, LikePhys is a training-free evaluation protocol that measures the model’s existing physics understanding without modifying its parameters. Concretely, LikePhys constructs controlled valid–invalid video pairs via simulation and uses the model’s denoising loss as a likelihood surrogate to read out an implicit preference over each pair. We then aggregate these preferences into the Plausibility Preference Error (PPE), which quantifies how often the model assigns higher likelihood to physically invalid videos. Unlike DPO, we do not introduce a reference policy, do not optimize a preference objective, and do not use human preference labels to fine-tune the model. Instead, PPE is designed purely as an evaluation metric that could, in principle, be combined with DPO-style training in future work, but is conceptually and practically distinct from it.
> >
> >
> > **Q6:If the model not only trained on realistic videos, how to adapt your work to evaluate this model?**
> >
> > **A6**: Our method is a general evaluation protocol for video diffusion models and does not make prior assumption on the training data distribution being realistic. In practice, the models we evaluate are also trained on diverse data that are not purely realistic. The idea is, while a model may have different responses to different visual styles due to training data, we do not use the absolute likelihood of single videos, but the likelihood difference within a controlled valid–invalid pair. As long as the two videos in a pair share the same visual appearance and differ only in physics adherence, the difference in estimated likelihood remains meaningful and can be attributed to the model’s physics understanding rather than its preference for a particular style.

---

### Author Response · Authors · 2025-11-20

Dear Reviewers and ACs,

Thank you very much for your insightful reviews and constructive comments, which help improve our manuscript. We have carefully taken all the suggestions and polished the submission, which has been updated in the final version. In the meantime, we identify a few common ambiguities and misunderstandings in our review and would like to clarify them below, along with the changes we made to better explain them in our updated manuscript.

1. Our method disentangles influence from visual appearance by comparing the relative likelihood of a controlled valid-invalid video sample pair rather than the absolute likelihood of a single sample. This is to ensure that the model-specific responses to visual appearance and other confounding factors cancel each other out in the pairwise comparison, and that the difference in the likelihood estimated for a pair arises solely from controlled physics violations. In this way, specific generative model bias to style should not influence the PPE metric, and the metric remains unbiased and comparable across models. We empirically verified this in Section 4.3 that the PPE metric is does not correlate to VBench visual quality metrics. We have revised Sections 1, Introduction and 3, Methodology, to clarify the method and remove potential ambiguity.

2. Our method does not rely on the assumption that the video diffusion models are trained on realistic videos. Our method aims to assess whether the learned video generative model distributions are close to physics-plausible distributions by comparing the likelihoods the model assigns to physically valid and invalid video pairs. Training data can be one of the factors influencing such physics understanding capacity; however, we do not make any prior assumptions about it. We have revised the discussion in Section 5 in the updated manuscript to remove potential ambiguity.


We would like to kindly ask reviewers if you have further questions, or if you find the updated manuscript and rebuttal resolve your question, we kindly ask you to update your rating accordingly.


Best Wishes,

All Paper 5897 Authors

---

### Meta-Review · Area_Chair_RH3L · 2026-01-02

**Summary:**

Reviewers raised concerns primarily about the fairness of comparisons between models trained on different data distributions, the reliance on simplified simulated scenes rather than real-world videos, and several clarification questions regarding the definition and computation of PPE. In the rebuttal, the authors provided detailed explanations and additional experimental evidence clarifying the violation-of-expectation formulation of PPE, its applicability to real-world video evaluation, and the robustness of the results across settings. These responses satisfactorily address the major concerns raised by reviewers, supporting a positive recommendation.

**Reviewer Concerns:**

### Addressed
* **Fair comparison between models trained on different data distribution**: The authors explained how PPE is calculated follows the violation-of-expectation paradigm, which should isolate other factors and only focus on differences arise from controlled violations of the governing physics dynamics.

* **Simple simulated textures or scenes than real-world videos**: The authors clarified that the impossibility of creating physically-implausible data in real world, and that the focus is on physical rules other than visual vias. The human study shows that PPE results on simulated data well aligns with human evaluation on real-world generations.

### Outstanding
* **Requirement of model access**. The approach requires access to the model, which is an inherent limitation. However, this does not substantially diminish the contribution of the paper. As noted by the authors, this requirement is relatively mild in the open-source setting, where open-source video diffusion models are widely used as research baselines. For closed-source models, the proposed method remains applicable from the model provider's perspective.

**Reviewer Scores:**

**Reviewer hTWK (initial score: 4)**. The reviewer raised several clarification questions while acknowledging that PPE aligns better with human evaluation than existing metrics and that it provides a valuable evaluation framework. The rebuttal effectively clarifies the reviewer's misunderstanding regarding the application to real-world videos, as well as the other raised points. Therefore, I would expect the reviewer to increase their score after discussion.

**Reviewer Bi5p (initial score: 4)**. The reviewer raised several questions, including (1) the applicability to closed-source video diffusion models, (2) the fairness of comparisons between models pretrained extensively on simulated data and those that are not, (3) the motivation of the proposed approach, and (4) the choice of timesteps. In the rebuttal, the authors addressed these questions with detailed clarifications and additional experiments. Therefore, I would expect the reviewer to increase their score after discussion.

**Reviewer tiXF (initial scores: 2)**. This reviewer raised concerns about a mismatch between assumptions and the actual computation of PPE, as well as distribution discrepancies between benchmark videos and evaluated models. These concerns were addressed in the rebuttal. Given the initially low score, I would expect the reviewer to maintain or potentially increase their score.

**Reviewer y5Dy (initial scores: 6)**. The reviewer submitted their response prior to the ICLR data leak and explicitly acknowledged that the authors had addressed most of their concerns, stating that they would maintain their positive rating in support of the paper.

---

### Decision · Program_Chairs · 2026-01-26

Accept (Poster)